# Cross-species dissection of saline-related genes by genetically deciphering a euryhaline microalga *Chlorella* sp

Aoqi Wang[1,5], Qinhua Gan[1,2,5], Yi Xin[1,5], Ying Deng[1], Xiao Han[1] & Yandu Lu [1,3,4] ✉

Deciphering adaptation to habitat shifts across the salinity boundary necessitates investigation of "lost" and "acquired" saline genes. By assembling a telomere-to-telomere genome, we propose that the euryhaline Chlorophyta *Chlorella* sp. MEM25 represents an early-diverging saltwater species that has evolved numerous genes essential for saltwater-freshwater transitions. By comparison with Viridiplantae genomes, we identify ancestral genes and lineage-specific genes related to salinity adaptation. Loss-of-function mutants of the proposed salt-sensitive genes in algae and plants exhibit increased salt resistance, highlighting the potential of the MEM25 genome as a breeding resource. Notably, the gene *RMI1* plays an important role in salinity tolerance across species, from microalgae to higher plants.

Life on Earth is thought to have originated 4 billion years ago in primordial, hypersaline oceans[1] with salinities of ~80 g·L$^{-1}$ [2]. Marine-freshwater transitions and terrestrialization (the transition from water to a terrestrial environment) are generally regarded as pivotal events in the evolution and diversification of land plant flora[3]. The transition to a terrestrial environment by plants has been documented frequently[4–7]; however, habitat shifts from marine to freshwater have not been thoroughly investigated. One prevailing notion is that the transition from salty to freshwater habitats occurred in the common ancestor of Chlorophyta (commonly known as green algae) and the conquest of the land started with Streptophyta (embryophytes and their closest algal relatives, known as streptophyte algae)[4,7,8].

Higher marine-freshwater transition rates than anticipated have, however, been reported for both prokaryotes[9] and eukaryotes (e.g., microbial eukaryotes[10]). Evolutionary studies based on long-read metabarcoding across the tree of eukaryotes reveals that such habitat transitions have occurred in both directions in almost all major eukaryotic lineages[10]. Large-scale genome sequencing defines the extent of protein-coding and viral elements in microalgal genomes and posits a unified adaptive strategy for algal halotolerance[11]. Increasingly, evidence supports the suggestion that, differing from terrestrialization, 'true land plants' evolved uniquely within the streptophytes, and

marine-freshwater transitions are prone to be lineage-specific and bidirectional[4,12–14].

To shed light on the genetic and cellular adaptations that have allowed crossing over the salinity barrier, the generation of genomic data from closely related marine and freshwater lineages is crucial[10]. An appropriate choice of organism group is essential for such studies. Marine-type sister groups of Streptophyta have not been confirmed[14]. Therefore, it is essential to find a speciose group with ample genome data available, occurring in both freshwater and saline water.

Chlorophyta microalgae of the genus *Chlorella* seem to satisfy these criteria. Their minute cell size and resistance to environmental stress facilitate long-distance dispersal. The genus is omnipresent in terrestrial and aquatic habitats[15]. Although the majority inhabit freshwater[16], a large number of species occur in the oceans, from the Antarctic, to temperate and tropical regions[17]. However, the genomes of saltwater *Chlorella* spp. have not previously been sequenced. The euryhaline *Chlorella* sp. MEM25 (hereafter MEM25) was originally isolated from the inland saline water of Hainan Island, China, in August 2016[18]. The water is highly saline (> 65‰), with a year-round high temperature (from 30 to 41 °C). MEM25 grows vigorously across a broad range of environmental conditions, including in salinities

[1]Engineering and Research Center of Marine Bioactives & Bioproducts of Hainan Province, School of Marine Biology and Fisheries, Hainan University, Haikou, China. [2]National Key Laboratory of Tropical Crop Breeding, School of Tropical Agriculture and Forestry, Hainan University, Haikou, China. [3]Haikou Innovation Center for Research and Utilization of Algal Bioresources, Haikou, China. [4]Key Laboratory of Tropical Hydrobiotechnology of Hainan Province, Haikou, China. [5]These authors contributed equally: Aoqi Wang, Qinhua Gan, Yi Xin. ✉e-mail: ydlu@hainanu.edu.cn

ranging from 0 to 105‰. With increasing salinity, maximum biomass production was observed at 70‰[18].

Here, we present genome sequences of MEM25, which represents one of the early divergences in the genus. The genome of MEM25, likely positioned at one of the bifurcation points separating saltwater and freshwater species, may reveal the suite of traits that facilitated saline adaptation. The high-quality genome assembly of MEM25 approaches the gold standard telomere-to-telomere (T2T) assembly through a meticulous blend of sequencing technologies, ensuring exceptional coverage and precision. By deciphering multi-omics data via integrated strategies, we propose that the genetic products of ancestral genes and lineage-specific genes collaboratively sculpt the adaptation of MEM25 to salinity changes. Comparison of MEM25 with freshwater green algae and halophilic *Dunaliella salina* underscores the intricate evolutionary dynamics entailed in transitioning between saltwater and freshwater ecosystems. With the recently established genome editing technologies[19,20], it allows us to outline a list of fundamental genes related to salinity that offer potential for enhancing saline tolerance of crops.

These findings allow us to address two questions important when exploring the evolutionary history of habitat transitions. (1) How did freshwater species evolve from their saltwater ancestors, or vice versa? (2) What gene repertoires contribute to the euryhaline characteristics of MEM25?

## Results

### Highly contiguous genome assembly

A hierarchical assembly approach yielded a high-quality nuclear genome for MEM25 of 53,506,137 bp, distributed over 16 chromosomes (Fig. 1a), as confirmed by 4′, 6-diamidino-2-phenylindole (DAPI) staining (Fig. S1). A high level of genome scaffold continuity was revealed by the long scaffold N50 (longest, 4.3 Mb; Supplementary Data 1) and a high value of Long Terminal Repeat Assembly Index (LAI) (15.33; Fig. S2), meeting the quality benchmark for reference genomes (i.e., between 10 and 20) (Supplementary Data 2)[21]. Notably, all sixteen

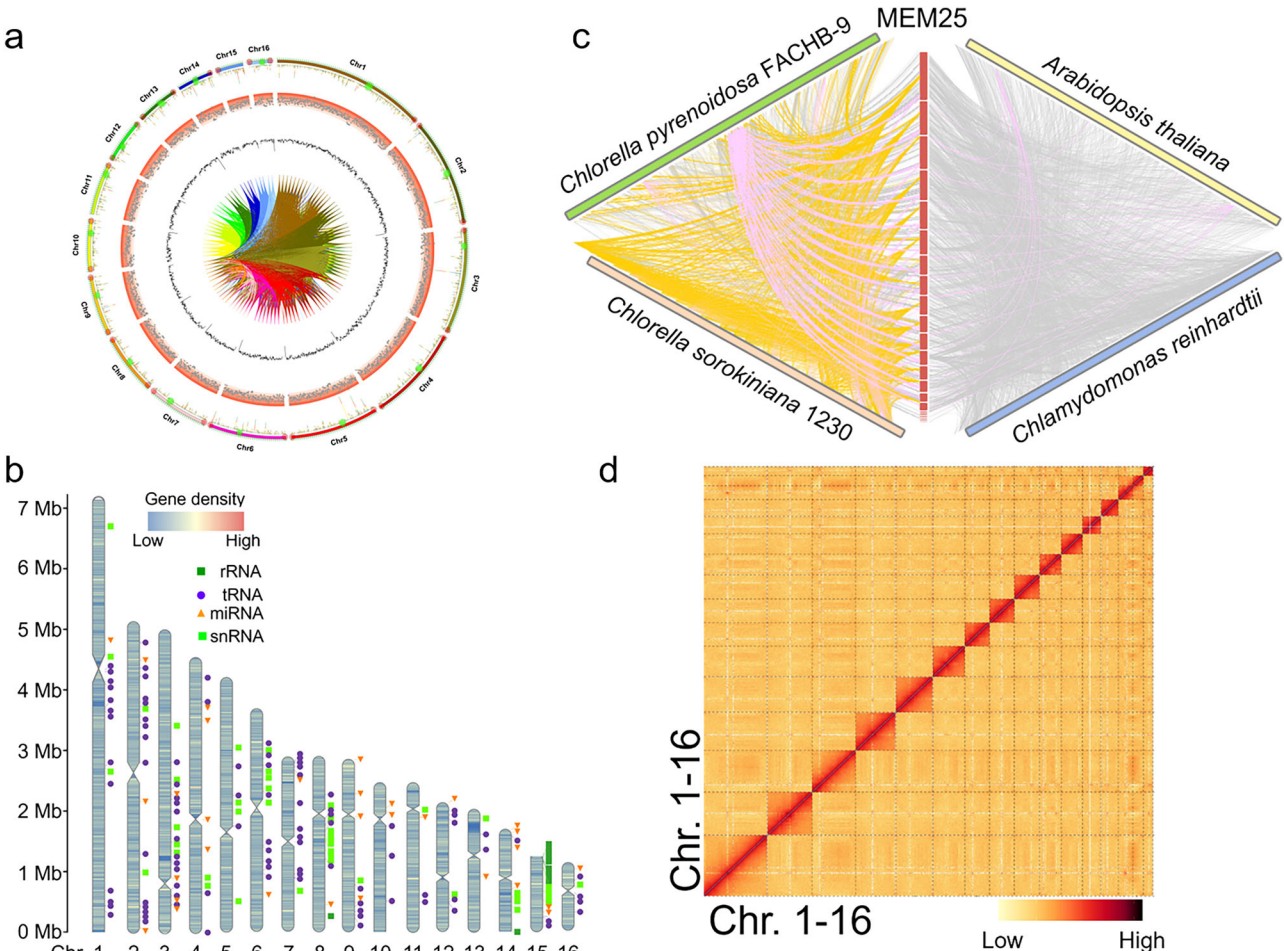

**Fig. 1 | De novo genome assembly of *Chlorella* sp. MEM25. a** Overview of the de novo genome assembly and sequencing analysis of MEM25. The concentric circles, from outermost to innermost, represent chromosome karyotype, gene density, GC content, and Hi-C interactions among chromosomes. All chromosome scales and distributions are drawn in a window size of 20 kb. For the outermost chromosome karyotype, the telomere and centromere are indicated by light red and light green circles, respectively. The bar chart displays the transcriptional changes under varying salinity conditions, with blue and yellow indicating low salt treatment for 3 hours, and orange and green representing high salt treatment for 24 hours. **b** The assembled sequence of the 16 chromosomes of the nuclear genome. The nominal plus strands progress from 5′ to 3′ left to right, displaying rRNA, tRNA, miRNA, snRNA, and gene density. The rounded corners at the chromosome edges signify the telomeres, while the twists represent the centromere regions. **c** Comparison of synteny between the MEM25 genome and genomes of chosen species. Collinearities in gene sequences, telomeres, and centromeres are delineated by the gray, purple, and yellow lines. The apparent synteny of MEM25 telomeres to a single region in the FACHB-9 genome results from the latter's fragmented assembly (1336 contigs). **d** Intensity signal heat map of Hi-C chromosomes. The line color from light yellow to dark red indicates the escalation in the interaction intensity. All source data are provided as a Source Data file.

chromosomes were clearly marked with centromeres (with three featured sequences; FSs in Fig. S3) and the plant-specific telomeric repeats (TTTAGGG, Supplementary Data 3; with twelve chromosomes containing telomeric repeats at both ends and Chr15 identified as a telocentric chromosome) (Fig. 1b).

An investigation into intragenomic synteny among the sixteen chromosomes of MEM25 (Fig. S4) and among three *Chlorella* genomes (MEM25, *Chlorella pyrenoidosa* FACHB-9[22], and *Chlorella sorokiniana* 1230[23]), along with the genomes of *Chlamydomonas reinhardtii*[24] and *Arabidopsis thaliana*[25] (Fig. 1c), revealed no whole-genome duplications (WGDs) in the MEM25 genome. Despite the fragmented genome assembly of *C. pyrenoidosa* FACHB-9 (hereafter FACHB-9) and *C. sorokiniana* 1230, better synteny was observed between MEM25 and these strains than that between MEM25 and *C. reinhardtii* or *A. thaliana*, suggesting the sequence conservation of centromeres and telomeres in chlorella strains (Fig. 1c).

A 100 kb-resolution 3D map of the MEM25 genome (Fig. S5) and whole-chromosomal interactions (Fig. 1d) were created using HiC-GNN and visualized using UCSF Chimera. Altogether, the quality, accuracy, and completeness of the MEM25 genome assembly is higher than the available reference sequences[22,26], ensuring the reliability of subsequent genomic analyses (See Supplementary note 1 for the details of genome assembly).

## MEM25 appears closest to one of the branch points separating saltwater and freshwater Chlorophyta

To illustrate evolutionary transition and distribution of saline genes, comparative analyses were performed on the genomes of MEM25 and 37 species of Chlorophyta, divided into marine ($n = 13$) and freshwater species ($n = 24$) (Supplementary Data 4). Representative species from cyanobacteria ($n = 2$; one saltwater and one freshwater), Rhodophyta ($n = 1$; saltwater), Glaucophyta ($n = 1$; freshwater), streptophyte algae ($n = 4$; freshwater), and embryophytes ($n = 2$) were used as outgroups. An ancient freshwater lineage *Chlorokybus atmophyticus* (a transitional species from streptophyte algae to land plants) and a halotolerant marine species *Picochlorum* spp (isolated from a brackish water pond subject to significant salinity fluctuations) were included to highlight critical transitions from green alga to land plants and from salt water to fresh water. Cyanobacteria (i.e., *Synechococcus moorigangaii* and *Synechocystis* sp. PCC 6803) were also included to provide a holistic view of genetic diversity in microalgal evolution. For the selected species, habitat information and genomic sources were thoroughly verified (See Supplementary Data 4 for details). Encompassing a diverse array of algae, spanning phylogenetically distant taxonomies, allowed for precise delineation of homologous gene families and provides insights into genetic diversity across microalgae.

To mitigate discrepancies stemming from nucleotide or transcriptional variations between prokaryotic and eukaryotic organisms, protein sequences were utilized for homologous gene family identification. Only 13 orthologs of single-copy genes were detected in the chosen Chlorophyta species (Supplementary Data 5). We thus constructed the species tree using the most closely related genes within single-copy or multi-copy orthogroups ($n = 199$), employing the Species Tree from All Genes (STAG) algorithm[27] (Fig. 2; see Fig. S6a for the tree of the chosen species with 1000 bootstrap replicates with bootstrap values above 70%). The highly credible species trees (bootstrap support exceeding 60%) align with the chloroplast phylogenetic tree (Fig. S6b), positioning MEM25 closest to the branch point separating freshwater *Chlorella* spp. from the marine species (Fig. 2).

The molecular clock places the emergence of MEM25 at 632 Ma, predating all examined *Chlorella* spp. and ranking among the oldest Chlorophyta (Fig. S7). Although certain clades diverged earlier than MEM25, their putative saltwater ancestors remain unidentified,

suggesting a dispersed and independent evolutionary transition from saltwater to freshwater habitats. Unlike the swift and dramatic transition to a terrestrial environment spurred by abrupt climate shifts, the transition from saltwater to freshwater settings unfolded in localized regions, with distinct lineages evolving independently within specific niches.

To mitigate potential biases from root placement or sampling constraints, we reassessed the phylogenetic connections of MEM25 within Chlorophyta exclusively ($n = 93$). To expand our cohort, we incorporated species from the 1KP project with transcriptomic data[8] and those with available genomes (Supplementary Data 6). This allowed for a better representation of Core Chlorophyta diversity, encompassing groups like Pedinophyceae, Chlorodendrophyceae, Trebouxiophyceae, Ulvophyceae, and Chlorophyceae. The unrooted tree affirmed the close evolutionary proximity of MEM25 to early marine eukaryotic microalgae, underscoring the disparities in evolutionary position and preference for saline growth conditions compared to other selected *Chlorella* strains (Fig. S8). While numerous green algae genera encompass both marine and freshwater species, *Chlorella* stands out as one of them. MEM25 potentially retains genes and traits associated with marine environmental adaptation, warranting further exploration into salt-adaptation mechanisms.

## MEM25 shares both saltwater- and freshwater-featured genes

To identify the distinctive gene families of freshwater and saltwater Chlorophyta species (Supplementary Data 4), we initiated with a two-stage machine learning approach involving 86 models and model combinations. Specifically, we generated a presence-absence matrix for gene families and applied various machine learning algorithms to differentiate between freshwater and saltwater algae (see "**Methods**"). The Random Forest + Elastic Net [0.6], identified as the most effective combination among the 86 models, achieved a classification accuracy of 97% and pinpointed 138 critical gene families with distinct functions in freshwater versus saltwater microalgae (Supplementary Data 7). These genes were further categorized into saltwater ($n = 44$) and freshwater-specific groups ($n = 94$) based on their weight coefficients.

A three-dimensional PCA analysis of these genes was conducted where freshwater species clustered tightly together and MEM25 emerged as the closest saltwater species to the freshwater cluster (Fig. 3). MEM25 displays characteristics of both saltwater (housing 32 of the 44 featured saltwater genes) and freshwater (possessing 75 of the 94 featured freshwater genes) adaptation. In comparison, the genome of *Chloropicon primus* (a prevalent species in marine phytoplankton communities[28]) contains 23 saltwater-featured genes and 34 freshwater-featured genes, respectively. The findings were reinforced by a more straightforward cutoff method, using a 60% cutoff (a gene family was categorized as a saltwater gene family if it was present in over 60% of saltwater species and absent in over 60% of freshwater species, and vice versa). A number of 614 featured gene families enriched in either saltwater ($n = 129$) or freshwater species ($n = 485$) (Supplementary Data 8 and Fig. S9a). Pearson's chi-square ($\chi^2$) tests confirmed the statistical significance (Pearson's $\chi^2$ (df=1) = 1430.5, $P = 2.2e{-}16$, Cramer's V = 0.49, 95% confidence intervals = [0.44, 0.54]), and similar results were obtained when we raised the cutoff from 60% to 80% (Fig. S9b).

This suggests that MEM25, potentially as an early-diverging saltwater species, has evolved numerous genes critical for freshwater-saltwater transitions, positioning it at one of the many pivotal junctures separating saltwater and freshwater Chlorophyta. The genome exploration of MEM25 could serve as a foundational element in unraveling the underlying molecular mechanism and processes involved in the transition in either direction between marine and freshwater.

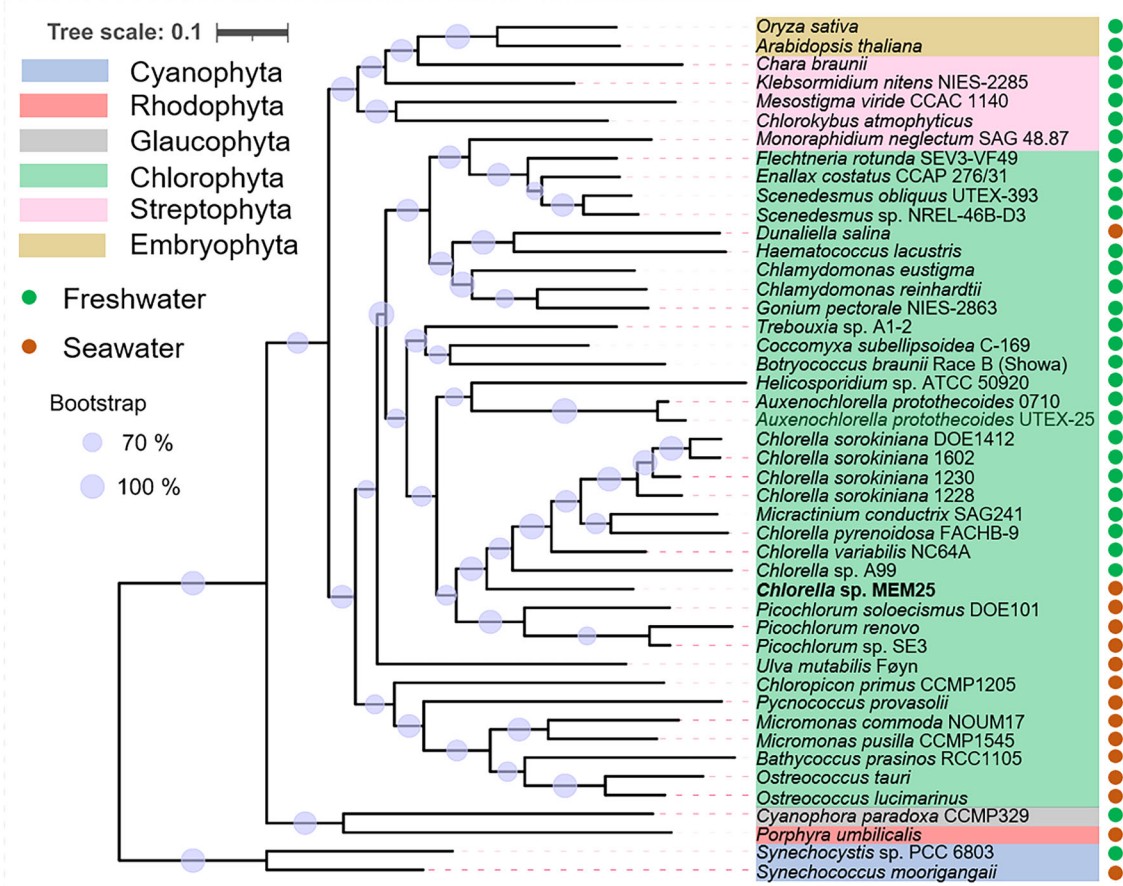

**Fig. 2 | Phylogenetic position of *Chlorella* sp. MEM25.** Rooted phylogenetic tree featured 38 Chlorophyta species and representative species from cyanobacteria, Rhodophyta, Glaucophyta, streptophyte algae, and embryophytes. The species tree was deduced using the most closely related genes within single-copy or multi-copy orthogroups ($n = 199$), based on the Species Tree from All Genes (STAG) algorithm. Species from different phyla are indicated in different background colors. Freshwater and seawater species are shown in green and orange dots. Bootstrap with values of 100% or above 70% are indicated in bigger or smaller dots at the branching points, respectively. All source data are provided as a Source Data file.

## MEM25 possesses general saline-related modules shared with freshwater species and exclusive ones that have not yet been explored

To investigate species similarities and differences in salinity adaptation mechanisms between saltwater and freshwater species, we conducted a comparative analysis of the global trancriptomic and metabolic dynamics of the euryhaline MEM25 and the neutrophilic freshwater FACHB-9[22] (See "**Method**" for growth conditions of FACHB-9). Despite the relatively high synteny observed in their genome sequences (Fig. 4a; see statistical details in Supplementary Data 9), there was a remarkably low alignment ratio of FACHB-9 transcriptomic reads on MEM25 genome sequences (0.95%), and vice versa (0.02%) (Supplementary Data 10). To mitigate biases stemming from differences in gene annotation quality between the two species, we employed a Weighted Correlation Network Analysis (WGCNA). By aggregating the expression of genes within each orthogroup to create a "metagene" expression level, we aimed to capture co-expression patterns reflective of shared functional roles, irrespective of individual gene copy numbers within each gene family.

We established a common set of annotations for 5,026 orthologous gene families, comprising 7,594 and 8,360 protein-coding genes, for comparisons between MEM25 and FACHB-9, respectively (Fig. 4b). By applying WGCNA on all 24 RNA-Seq datasets, we identified 67 gene modules (shown in various colors in Fig. 4c; |Pearson| ≥ 0.75 and $P \leq 0.005$), with gene numbers ranging from 31 to 233. Subsequently, we assessed the associations of each module with four parameters:

stress severity (salinity), stress duration (treatment duration), damage state (0 for algae under preferred salinity, 1 if not), and MEM25-specific mechanisms (0 if not specific to MEM25, 1 if specific to MEM25) (Supplementary Data 11). Notably, none of the modules exhibited correlations with stress duration, while 15 modules displayed significant associations with salinity, either positively ($n = 3$) or negatively ($n = 12$). Additionally, we identified 20 modules specifically linked to freshwater species and 13 modules associated with saltwater species, encompassing 1,841 and 1,053 orthologs, respectively (Fig. 4c).

Specifically, we found that the thistle2 module was significantly (Pearson correlation = 0.89, df = 45, $P = 9 \times 10^{-9}$, 95% confidence intervals= [0.80, 0.94] $P = 9 \times 10^{-9}$, Pearson = 0.89) and specifically associated with saline adaptation of MEM25, suggesting that the feature of this subfamily was lost in the lineage leading to the freshwater counterpart. Noteworthy is the distinct association of the RecQ-mediated genome instability protein 1 (RMI1; CP5g5156) with the saline adaptation of MEM25 (Supplementary Data 12). Nonetheless, the transcription of *RMI1* notably decreased in both strains upon the transition from low to high salinity levels (Supplementary Data 12), indicating *RMI1*'s involvement in responding to heightened salinity, likely following a similar pattern in both freshwater and saltwater *Chlorella* species. The apparent inconsistencies could be elucidated by the considerably lower abundance of *RMI1* transcripts in MEM25 relative to FACHB-9 (Supplementary Data 12). A relatively diminished dosage of *RMI1* appears to facilitate MEM25's acclimatization to saltwater. On the other hand, 69 genes of the 47 orthogroup in this

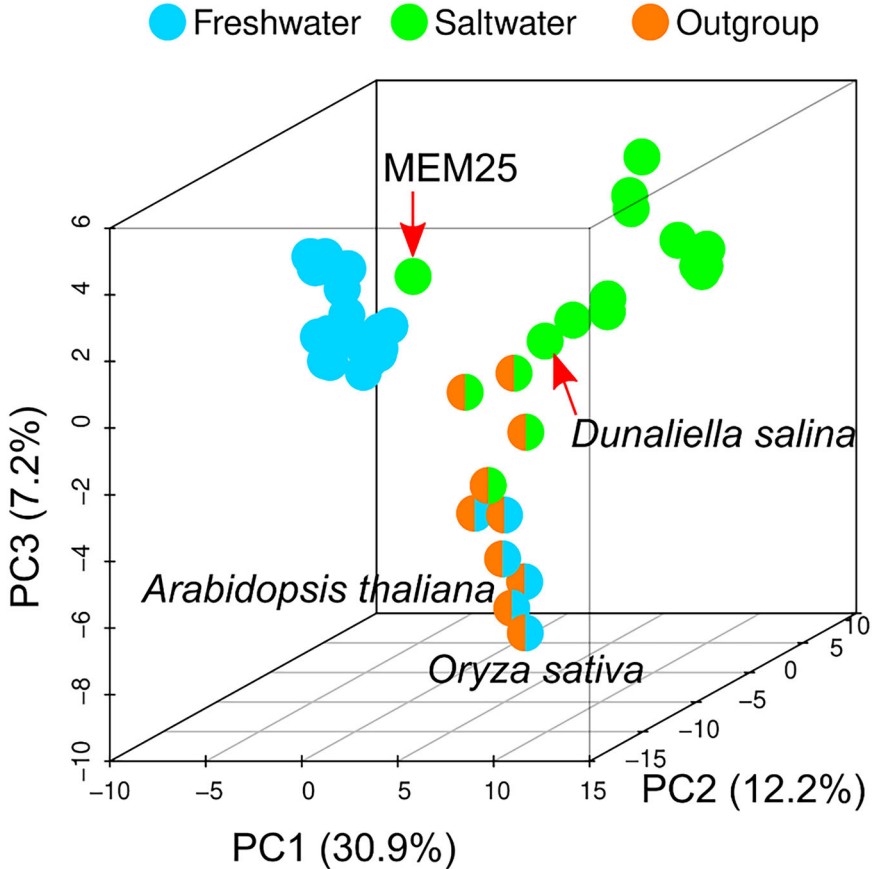

**Fig. 3 | MEM25 shares both saltwater- and freshwater-featured genes.** Clustering of the 36 Chlorophyta species is based on a multiple machine learning approach. When PCA was performed, no assumptions about the data were made, nor were the principal components predetermined. PC1 explained 30.9% of the variance, primarily reflecting a gradient of ecological adaptation to salinity across species; PC2 accounted for 12.2% of the variance, capturing variation in physiological traits within specific clades; PC3, which explained 7.2% of the variance, separated marine species from freshwater species, highlighting distinct evolutionary adaptations. MEM25 and halophilic *Dunaliella salina* are indicated. *Arabidopsis thaliana* and *Oryza sativa* are included to show the major evolutionary divergence between the outgroup and green algae (marine and freshwater species). This labeling helps validate the PCA results, ensuring that the distances between the outgroup and the study subjects accurately reflect their known evolutionary relationships. All source data are provided as a Source Data file.

module were largely unannotated (Supplementary Data 12), indicating potential salinity adaptation strategies unique to MEM25.

The metabolomic counterparts to the thistle2 module in the transcriptomic comparison were the black and pink modules (Fig. 4d). While most genes in the thistle2 module (Fig. 4c; specifically linked to MEM25's salinity adaptation) lacked annotation, the black and pink modules in the metabolome analysis contained known osmoprotectants like D-sorbitol, D-arabitol, and glycerolipids (Supplementary Data 13). Nevertheless, the roles of most other genes and metabolites in these modules in salinity adaptation remained unexplored. The presence of lineage-specific modules may suggest specific adaptations of saltwater or freshwater *Chlorella* sp. to saline stresses. The identification of MEM-specific or general modules of both freshwater and saltwater species could shed light on the molecular mechanism underlying transitions between salty and freshwater environments.

**Gains and expansions of saline-related genes are either exclusive to MEM25 or shared across Viridiplantae**

The homolog matrix of orthogroups was analyzed to deduce ancestral genes and lineage-specific genes content within the Viridiplantae (green plants), unveiling 5, 0, and 7 significantly expanded gene families in the genomes of Streptophytina, Chlorophyta, and *Chlorella* spp., respectively ($p < 0.05$; Fig. S7). For example, evolutionary analysis of gene family OG0000035 using the Computational Analysis of gene

Family Evolution (CAFE) framework yielded a family-wide $p$ value of 0.001, indicating significant deviation from the neutral birth–death model. The global evolutionary rate parameter ($\lambda$), which reflects the background rate of gene family expansion and contraction, had a mean value of 0.000414 (range: 0.0000613–0.0009377). To evaluate lineage-specific rate heterogeneity, we performed a Likelihood Ratio Test (LRT) comparing a branch-specific $\lambda$ model with the global $\lambda$ model (df = 9). Together with the Viterbi $p$ values generated by CAFE (ranging from 0.0005068 to 0.950976), the results demonstrate that Streptophytina lineages exhibit $\lambda$ values that significantly deviate from the global rate, indicating heterogeneous evolutionary dynamics within this clade. Comprehensive statistical outputs for additional gene families and species are provided in Supplementary Data 14.

Compared to all examined freshwater Chlorophyta, including *Chlorella* spp., a variety of gene families were expanded in the MEM25 genome (n = 89), implying their involvement in salt adaptation. Among these expansions, only five orthogroups have been found shared in green plants and could be referred to as lineage-specific genes. Out of these, four have been annotated and are associated with oxidative stress (i.e., peroxidase, family OG0015353), osmotic adjustment (i.e., volume-regulated anion channel subunit LRRC8A, family OG0021391; amino acid permease 2, family OG0026817), and response to salty stimuli (E3 ubiquitin-protein ligase, E3L; family OG0012359)[29–31] (Supplementary Data 15), suggesting these saline-related genes were

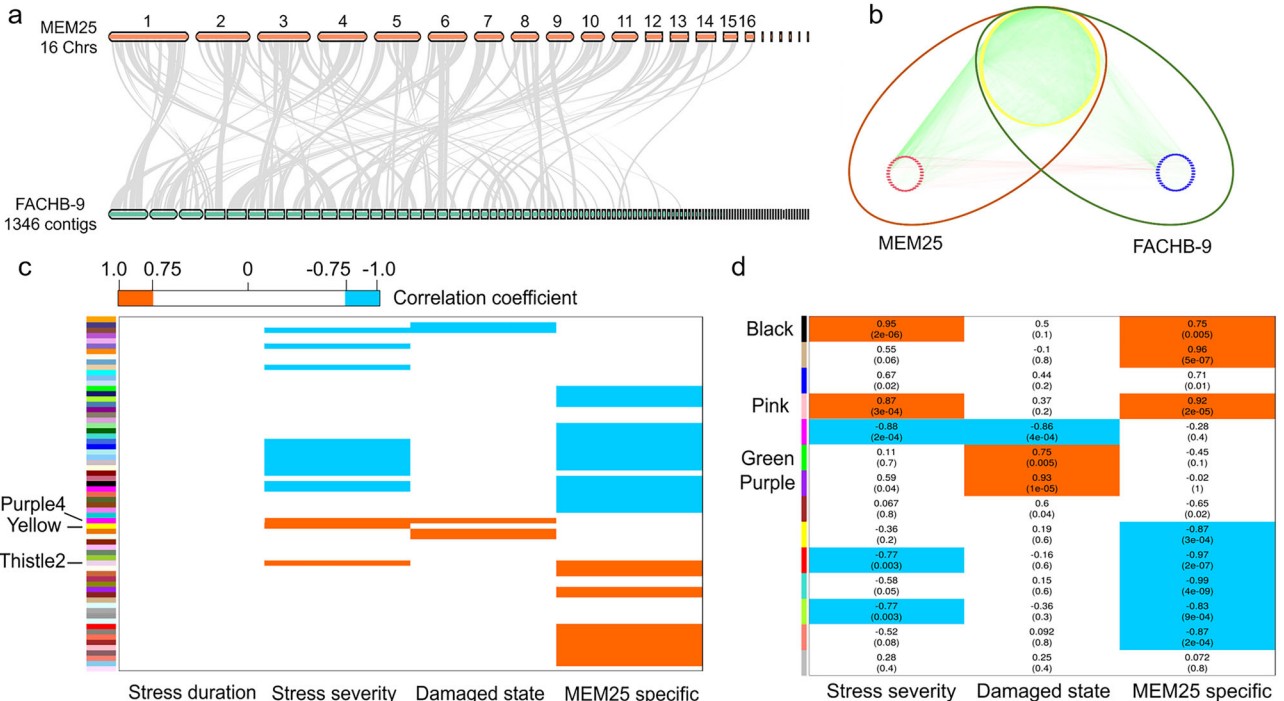

**Fig. 4 | Weighted gene co-expression network analysis (WGCNA). a** Genomic synteny comparison between *Chlorella* sp. MEM25 and *Chlorella pyrenoidosa* FACHB-9. **b** Protein-protein interaction analysis between MEM25 and FACHB-9. MEM25 and FACHB-9 proteins are depicted by the dark red and the dark green ellipses, respectively. Unique genes of MEM25 and FACHB-9 are shown as red and blue circles, while shared genes are represented by yellow circles. Lines connecting the circles signify protein interactions. **c** Module-trait relationships. The matrix plot displays correlations between modules (*Y* axis) and stress severity, stress duration, damage state, and

MEM25-specific mechanism. MEM25 and FACHB-9 have 5026 shared orthologs. The expression level of each ortholog is the sum of all genes within the family. The color scale indicates module-trait correlation from −1 (blue) to +1 (red), with cutoffs at ±0.75 ($P < 0.005$). See Supplementary Data 9 for the statistics details. **d** Metabolome comparison using WGCNA. MEM25 and FACHB-9 share 451 metabolites. The number outside brackets in a cell shows the Pearson's correlation $r$ value between a module and the parameter, while the number inside the brackets indicates the Pearson's correlation $p$ value. All source data are provided as a Source Data file.

already present in the last common ancestor of MEM25 and green plants prior to their divergence into either saltwater or freshwater preferred adaptations.

On the contrary, the majority of expanded groups are unique to MEM25 (84 in 89). We have designated these genes as lineage-specific genes of MEM25. The large number of MEM25-specific genes may reflect the phylogenetic distance of MEM25 from other *Chlorella* species with sequenced genomes. Assuming a freshwater origin of the ancestor of MEM25, portion of these genes may have been acquired during the specialization of MEM25 to a salty niche. For example, we found an exclusive presence of gene family OG0011211 in the MEM25 genome (Fig. S10). OG0011211 encodes a NAD-dependent epimerase/dehydratase family protein, absent in other Chlorophyta and plant lineages but found in three copies within MEM25 genomes (Fig. S10). This gene family is believed to have originated from bacteria where it is involved in osmotic stress[32]. Loss of NAD-dependent epimerase/dehydratase is known to cause osmosensitivity in bacteria[32], indicating that MEM25 may have acquired specialized adaptive mechanisms from bacteria through horizontal gene transfers (HGTs), representing an evolutionary innovation associated with salinity-related genes in MEM25.

For the remaining of MEM25-specific genes, we would assume them as orphan genes which have no apparent homology to genes in other evolutionary lineages occur in all genomes. They could be candidates for the de novo evolution of genes and may contribute to evolutionary novelties, which can become relevant for lineage-specific adaptations. Consequently, MEM25 has developed a collection of expanded salinity-related genes, categorized as ancestral genes and lineage-specific genes, shaping its euryhaline nature.

## Expanded genes in MEM25 are transcriptionally active in response to saline stresses

In order to unveil the molecular mechanisms of its euryhaline feature, we tracked the transcriptomic and metabolic dynamics of MEM25 under specific low and high salinity conditions. Differential gene expression analysis demonstrated that numerous genes which had undergone gene family contraction/expansion also displayed differential expression in response to changes in salinity. Among the 89 expanded orthogroups, 73% showed a significant correlation with salinity change (Fig. S11), suggesting an adaptation mechanism of MEM25 to saline environments at both genomic and transcriptional levels. For instance, all five annotated expanded orthogroups in MEM25 (i.e., family OG0012359, OG0015353, OG0011211, OG002139, OG0026817; Supplementary Data 15) were markedly upregulated following an increase in salinity (Fig. 5a). The transcriptionally enriched expanded genes also encompass those involved in fatty acid elongation (e.g., acyl-CoA thioeaterase encoding gene CP11g7958 and enoyl-[acyl-carrier-protein] reductase encoding gene CP15g9776) and antioxidant activity, such as glutathione metabolism (CP15g9764 and CP2g1989), ascorbate metabolism (L-galactose dehydrogenase CP8g6549 and monodehydroascorbate reductase CP1g297), vitamin C biosynthesis (GDP-D-mannose 3′, 5′-epimerase CP14g9420), alcohol dehydrogenase (CP9g7414), and inositol oxygenase (CP4g4403)[33] (Fig. 5b and Supplementary Data 16). Therefore, the expanded genes appear to actively transcribed in the saline adaptation of MEM25.

## Metabolic dynamics are synchronized with transcriptional changes

Regarding metabolism, the levels of 150 out of 451 metabolites underwent significant changes ($|\log_2(FC)| \geq 1$ and $P \leq 0.05$).

a

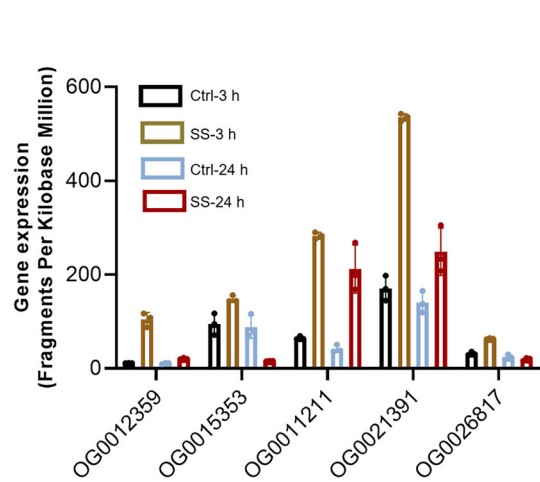

b

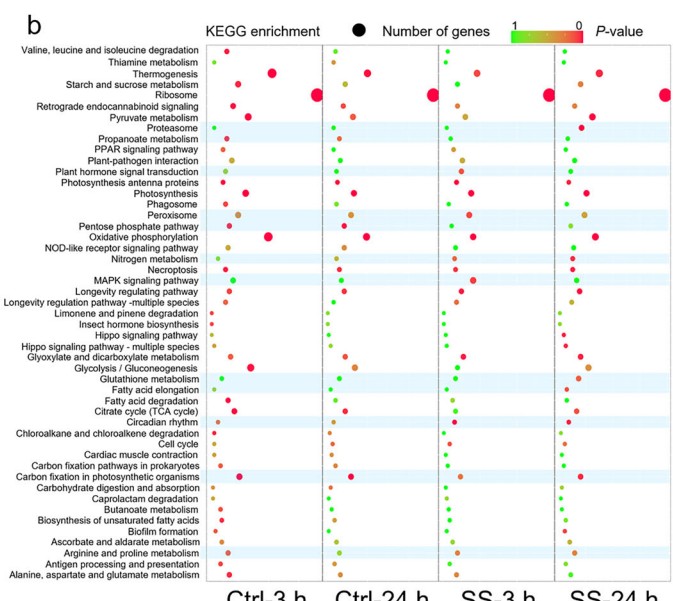

**Fig. 5 | Multi-omics integration of *Chlorella* sp. MEM25. a** Examination of the transcriptional dynamics of five annotated expanded orthogroups in MEM25 under the high-salinity condition. Data are presented as the means ± SD ($n = 3$). **b** KEGG pathway analysis performed on highly differentially expressed genes using the cluster Profiler package in R. A two-sided Fisher's exact test was applied, with the background set defined as all KEGG-annotated genes in the organism ($n = 3,833$). $p$ values were adjusted for multiple comparisons using the Benjamini-Hochberg (BH) method. Terms with a BH-adjusted $p$ value < 0.05 were considered statistically significant. The analyses were conducted without pre-filtering thresholds for gene set size or raw $p$ value, using a custom annotation to define term-gene associations. The $y$ axis displays pathway names, and the $x$ axis shows various treatments. The colored bar indicates the corrected $p$ value while green represents higher values and red represents lower values. Cyan blocks indicate pathways that have undergone significant changes in different treatments. Ctrl, 35‰ salinity; SS salt stress (105‰ salinity). All source data are provided as a Source Data file.

Corresponding to the upregulation of genes involved in proline biosynthesis (e.g., glutamate 5-kinase CP14g9503 and CP14g9505; delta-1-pyrroline-5-carboxylate synthetase CP8g6403; pyrroline-5-carboxylate reductase CP3g3123) and fatty acid desaturation (e.g., acyl-[acyl-carrier-protein] desaturase CP11g7888) (Supplementary Data 17), the contents of proline and unsaturated fatty acid significantly increased under high salinity conditions (Fig. S12 and Supplementary Data 18). We also noted elevated levels of metabolites with established roles in abiotic stress adaptation, including D- (+)-sucrose, gulonolactone, and vitamins (e.g., vitamin B13) (Supplementary Data 18). Following salinity changes, MEM25 tends to activate the biosynthesis of stress-related metabolites such as proline and sucrose, while the processes related to antioxidant activity were stimulated. Thus, the findings from genome expansion and omics analysis complement each other, revealing an array of saline-related genes unique to MEM25 or shared with Viridiplantae. These genes merit further validation to unveil their bona fide function and could potentially be leveraged for trait improvement in microalgae or crops.

### GWAS corroborates the salinity-related role of genes identified through genome expansion analysis

To validate the salinity-related genes uncovered through genome expansion, genome-wide association studies (GWAS) was conducted. An ethyl methane sulfonate (EMS) mutant population was generated using MEM25 as the parent strain. Out of 50,000 independent mutant lines, 536 mutants with altered $OD_{750}$ under either low (i.e., 35‰) or high (i.e., 70‰) salinity were selected for further phenotyping under the high-salinity conditions. Subsequently, 365 mutants with stable phenotypes after ten generations were chosen to assess variations in cell number, cell size, and dry weight (DW) under the high-salinity conditions (Fig. 6a). Lastly, the genomes of 195 saline-related mutants and three WT samples were resequenced with 30 × coverage ($n = 198$).

In total, 167,199 variations were detected in the genomes of all samples (Fig. 6b). Using the mixed linear model to analyze the

associations between the three phenotypes and genetic variations, we consistently pinpointed five loci exceeding a significant threshold ($- \log_{10} P \geq 5.0$), linked to either known or unknow genes. Specifically, we observed clusters of highly significant Single Nucleotide Polymorphisms (SNPs) proximal to S04_3657761 in Chr04 associated with the DW ($- \log_{10} P = 8.22$) (Fig. 6c; see the quantile-quantile plots in Fig. S13). S04_3657761 is positioned downstream of gene CP4g4326 and upstream of CP4g4327 with an intergenic region of 879 bp (Fig. 6d). The G to T transition at S04_3657761 creates a binding site for the transcriptional factor cysteine-rich polycomb-like protein which is associated with abiotic stresses, such as salinity, in a broad spectrum of species[34, 35]. Examination of the chromatin organization unveiled topologically associating domains (TADs) and loops displaying a significant interaction between S04_3657761 and gene CP4g4237 (Fig. 6e).

Intriguingly, all three genes (i.e., CP4g4326, CP4g4327, and CP4g4237) are *E3L*, supporting a salinity-related role of E3L, reinforcing its salinity-related role as indicated by genome expansion analysis and transcriptomic investigations (Supplementary Data 15). While the categories of *E3L* genes identified through GWAS, genome expansion, and transcriptomic studies may not perfectly align, they suggest a coordinated orchestration of E3L-mediated posttranslational modifications at both the genomic and the transcriptomic levels, contributing to MEM25's response to salinity. E3L has been demonstrated to play a crucial role in protein modification and the regulation of various biological processes and cellular responses to stress signals in plants[36]. Therefore, this association effectively pinpoints genes linked to functional polymorphisms, corroborating the findings of genome expansion and omics analyses.

### Functional validation reinforces the utility of the newly discovered salinity-related gene repertoire

To delve deeper into the salinity-related genes, an association network was established to show transcript-metabolite correlations in MEM25

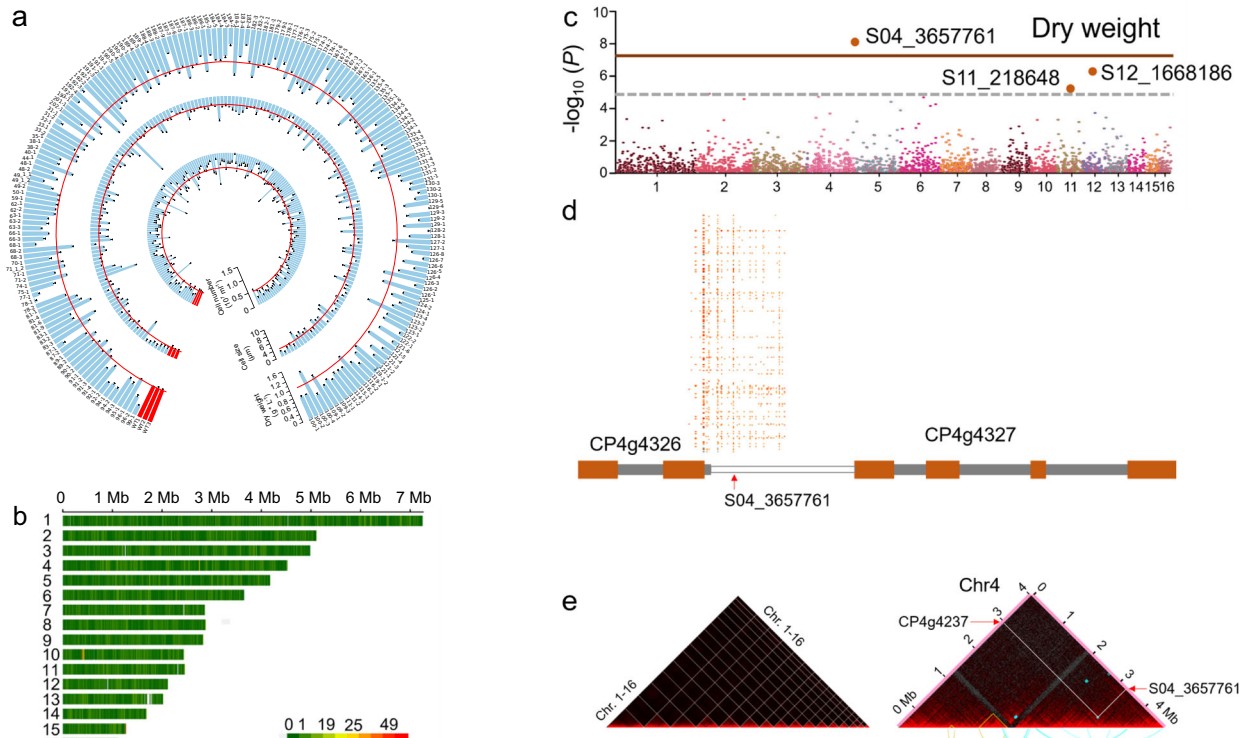

**Fig. 6 | Genome-wide association studies (GWAS) of *Chlorella* sp. MEM25.**
**a** Phenotype of the 195 MEM25 mutants. Microalgae were cultured into F2 medium (with a salinity of 70 g·L$^{-1}$) at an initial OD of 0.3, at 25 °C and 50 µmol·photons·m$^{-2}$·s$^{-1}$ light intensity. The dry weight, cell number, and cell size were determined after 10 days. The average values of each phenotype for the wild type are marked by the red lines. Data are presented as the means ± SD ($n = 3$). **b** Distribution of genetic variations ($n = 167,199$) across the 195 mutants' genomes. The color scale (bottom) indicates the density of variations per 1 kb, ranging from low (green) to high (red).
**c** Genome-wide association scan for dry weight using a linear mixed model implemented in GEMMA. The analysis cohort comprised an EMS-mutagenized population of 198 individuals, including 195 mutant lines and three wild-type controls. The $x$ axis indicates SNP locations along the 16 chromosomes, separated by vertical gray lines; the $y$ axis shows the $-\log_{10}$ ($P$ value) from each method. The upper red line and dash–dot line signify the significance threshold ($P = 5.98 \times 10^{-7}$)

and suggestive threshold ($P = 5 \times 10^{-5}$), respectively. The lead SNP S04_3657761 on chromosome 4 exhibited the strongest association with dry weight ($\beta = -0.01618 \pm 0.002495$, $t(196) = -6.483$, $P = 6.07 \times 10^{-9}$, 95% CI [ $-0.02107$, $-0.01129$]), remaining highly significant after multiple testing correction (Bonferroni-adjusted $P = 2.03 \times 10^{-5}$; FDR$_{BH} = 8.83 \times 10^{-7}$). **d** Location of the S04_3657761 region. This region is situated downstream of gene CP4g4326 and upstream of CP4g4327. Exons are indicated as orange boxes and introns are shown in gray boxes. Empty boxes are intergenic regions. Colored dots indicate the Single Nucleotide Polymorphisms (SNPs) proximal to S04_3657761 in Chr04. Arrow indicates the position of nucleotide variation in S04_3657761. **e** Mapping of topologically associating domains (TADs) and loops on chromosome 4. TADs are highlighted with yellow lines, while loops are depicted by blue lines. The interaction between S04_3657761 and gene CP4g4237 is illustrated by red arrows. All source data are provided as a Source Data file.

(Fig. 7a). This network included orthologous gene families ($n = 1231$) and metabolites ($n = 109$) that exhibited significant correlations with stress severity (i.e., salinity), with only pairs having a correlation coefficient above 90% considered in the subsequent analysis. Genes where both the transcript and the correlated metabolite displayed correlations with salinity changes were retained in the repertoire. To validate this repertoire, salt-sensitive genes were selected using three strategies. First, genes with elevated transcriptional levels (in FACHB-9 compared to MEM25) and also in the modules linked to freshwater species (not associated with the MEM25-specific mechanism) (e.g., CP4g4492 and CP1g58) were chosen. Second, genes with reduced transcriptional levels (under high salinity compared to low salinity conditions) and found in the modules associated with saltwater species (associated with the MEM25-specific mechanism) (e.g., CP5g5156 and CP8g6201) were selected. Third, genes from expanded orthologs in freshwater species (in comparison to MEM25) and present in modules associated with freshwater species (not linked to MEM25-specific mechanisms) (e.g., CP11g8082 and CP10g8454) (Supplementary Data 19) were used.

A gene inventory (including genes largely unannotated) was compiled based on these strategies and we expected improved salt

tolerance by knocking out individual genes in the repertoire. The homologs of five chosen genes have not been functionally investigated while one gene has been linked to DNA repair and meiosis in plants[37]. None of them have ever been associated with salinity tolerance.

MEM25 thrives best at 70‰ salinity, while FACHB-9 prefers 0‰ salinity. Recognizing the challenge of conferring salinity resistance to a freshwater species through single gene manipulation, we utilized a "buffering" strain, *Nannochloropsis oceanica*, adapted to a "medium" salinity level of 35‰ to evaluate the biological significance of the selected genes. Knockout mutants of all six candidate genes were created for *N. oceanica*. In comparison to the wild type, all mutants exhibited significantly increased biomass (Fig. 7b) and cell numbers (Fig. 7c) under the high salinity conditions (i.e., 70‰).

The successful validation of gene function has inspired us to explore the potential application of these genes for trait improvement of higher plants. *CP1g58* of MEM25 encodes a protein homologous to the *Arabidopsis* zinc knuckle (CCHC-type) family protein (AT3G42860), which has not been experimentally validated. On the other hand, the *Arabidopsis* counterpart of the protein encoded by *Chlorella CP5g5156* is RMI1 (AT5G63540), with the *Arabidopsis* mutant *rmi1* associated with DNA damage repair[37]. *CP1g58* in MEM25 (Fig. S14a)

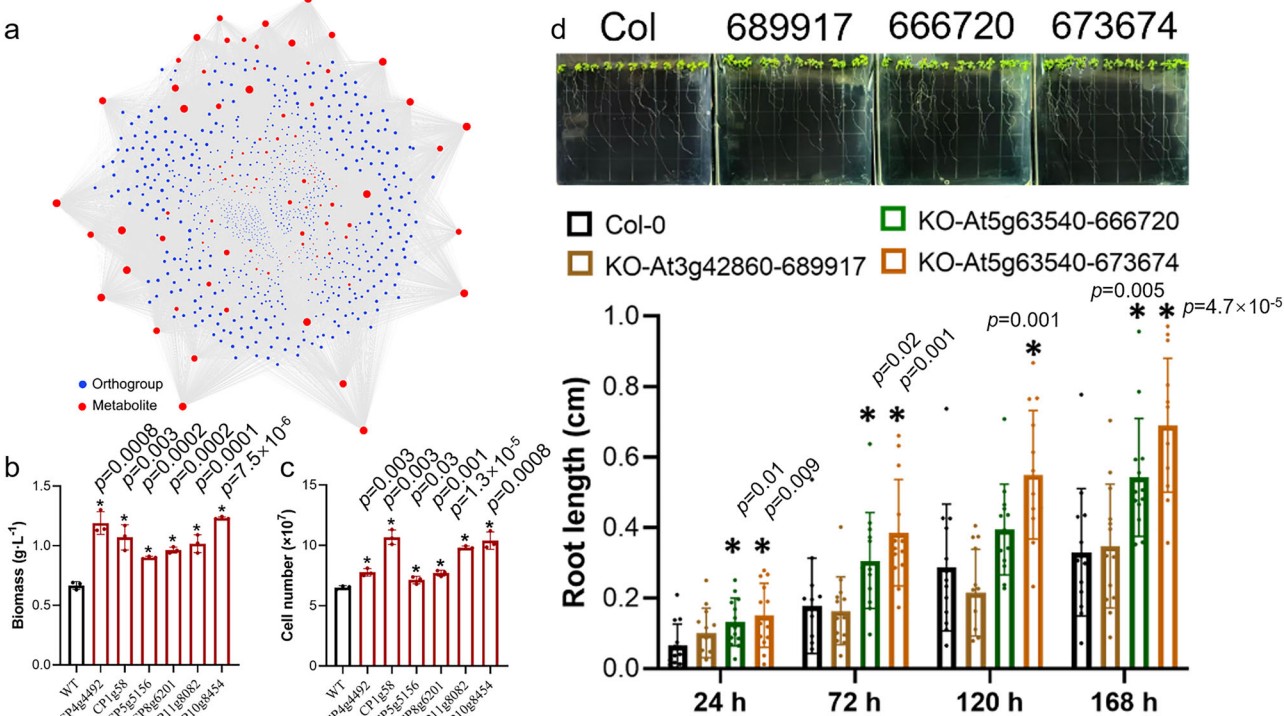

**Fig. 7 | Functional validation of newly discovered salinity-related genes.**
**a** Transcript-metabolite correlations. A total of 1,231 orthologous gene families and 109 metabolites exhibiting significant correlations with changes in salinity (correlation coefficient > 0.9) are included. Metabolites and orthogroups are represented by red dots and blue dots, respectively. **b** Biomass production of *N. oceanica* mutants. **c** Cell density of *N. oceanica* mutants. The *N. oceanica* equivalents of CP4g4492, CP1g58, CP5g5156, CP8g6201, CP11g8082, and CP10g8454 are

No7.g304, No80.g2678, No267.g7535, No185.g5882, No295.g8628, and No66.g2227, respectively. **d** Root length in the knockout mutant of the *Arabidopsis* homologs (AT3G42860 and AT5G63540) under a 150 mM salt concentration. Two independent knockout plant lines were examined for AT5G63540 (i.e., 666720 and 673674). Data are presented as the means ± SD ($n = 3$ for **b**, **c**; $n = 13$ for **d**). Asterisks (*) indicate statistically significant differences ($P \leq 0.05$) with two-sided test. All source data are provided as a Source Data file.

and the homologous gene *g3721* in freshwater FACHB-9 (Fig. S14b) exhibited significantly decreased expression levels under the elevated salinity. For *CP5g5156* in MEM25 (Fig. S14c), two homologous genes were identified in FACHB-9 (g1423 and g1502), all showing decreased expression levels with increasing salinity (Fig. S14d). These findings suggest that lower expression of *CP1g58* and *CP5g5156* homologs in *Arabidopsis* may confer enhanced resistance to salt. Thus, we choose these genes for further validation in higher plants.

In line with our expectations, two independent knockout plant lines of AT5G63540 (*RMI1*; homologs of *CP5g5156*) exhibited increased root growth under salt stress (Fig. 7d), indicating *RMI1* as a negative regulator in salinity tolerance in plants. *RMI1* may exemplify one of the numerous ancestral genes contributing to saline adaptation specific to Chlorophyta in freshwater or saltwater. In contrast, compared to the wild type, the knockout mutant of gene AT3G42860 in *Arabidopsis* (homologs of *CP1g58*) exhibited marginal difference in root growth under salt stress (Fig. 7d). Our findings suggest that the proteins encoded by AT5G63540 and its homologs play relevant roles in salinity tolerance across various species, from microalgae to plants. To our knowledge, this is the first report to document the effect of these genes on salinity adaptation, reinforcing the value of the gene repertoire identified here for pinpointing candidate genes linked to salinity resistance that could enhance plant agronomic traits.

## Discussion
The dynamic adaptation of green algae from saltwater to freshwater is a complex evolutionary process that showcases the diversity of selective pressures from marine and freshwater environments. Both saltwater and freshwater species have independently evolved or

inherited various adaptive mechanisms from their predecessors. MEM25 acts as a transitional point crossing over the salinity barrier, demonstrating dual environmental adaptation mechanisms. The genetic products of ancestral genes and lineage-specific genes collaboratively sculpt the adaptation of MEM25 to salinity changes. MEM25 possesses general modules related to salinity that are shared among freshwater Chlorophyta species (potentially extending to all green plants), alongside distinctive modules acquired through HGTs (e.g., OG0011211) or other yet undiscovered mechanisms (e.g., orphan genes).

In the hypothetical scenario of transitioning from saline to freshwater habitats, the shared modules may encompass a suite of genes (referred to as ancestral genes) inherited from saltwater forebears, equipping freshwater species with analogous, though largely mitigated, mechanisms to confront salinity fluctuations. These genes' functionalities revolve around mitigating or intensifying cellular harm in saline settings, representing a ubiquitous mechanism present in both saltwater and freshwater species. An example is that MEM25 employs proline as an osmoprotectant, commonly found in embryophytes. This indicates that the mechanism of employing proline as an osmoprotectant in higher plants may originate from their shared ancestral lineage with MEM25. Furthermore, as an early-diverging marine alga, MEM25 has commenced the traversal of the salinity threshold. While boasting numerous genes specific to saltwater environments, MEM25 is in the process of assimilating genes characteristic of freshwater milieus. MEM25 harbors distinctive modules (referred to as lineage-specific genes) that likely evolved autonomously, enhancing its ability to thrive in its unique saline habitat.

Alternatively, in a scenario where MEM25's ancestors originated in freshwater, the acquisition of lineage-specific genes transpired during the specialization phase towards an environment with higher salinity levels. For instance, the gene family OG0011211 (belonging to the NAD-dependent epimerase/dehydratase family) could have been assimilated from bacteria through HGTs. These lineage-specific genes embody genes associated with salinity that may have been "lost" in freshwater counterparts. The gradual disappearance of lineage-specific genes may have conferred a competitive edge for saltwater ancestors adapting to freshwater environments. In both conjectures, these genes likely facilitated MEM25's acclimatization to challenging habitats and warrant further investigation for enhancing resilience to salinity.

MEM25 and *D. salina* represent two typical adaptive strategies among euryhaline and halophilic green algae, respectively. What sets MEM25 apart is its ability to adapt to both freshwater and saltwater environments, maintaining its status as a dominant species in both ecological settings. The euryhaline nature of MEM25 enhances its competitiveness across a wide range of ecological environments, whereas the extreme salt tolerance specialization of *D. salina* limits its ecological adaptability. As a possibility, euryhalinity might be explained by sodium excretion rather than salt tolerance as seen in some euryhaline embryophytes[38]. It is apparently the case for halophilic *D. salina* while whether sodium excretion is involved in MEM25's euryhalinity remains to be explored. *D. salina*'s salt tolerance primarily relies on accumulating high concentrations of glycerol as an osmoprotectant to maintain cellular osmotic pressure, relying on efficient ion exclusion systems, and employing antioxidant enzymes to reduce reactive oxygen species[39]. However, many of these mechanisms are specialized for high-salinity environments, resulting in significantly reduced metabolic flexibility under low salinity or even freshwater conditions. In contrast, MEM25 exhibits a more diversified adaptability, with its salt tolerance mechanisms encompassing not just the accumulation of osmoprotectants but also long-term evolutionary mechanisms at the genomic level, mid-term adaptation mechanisms at the metabolomic level, and rapid response mechanisms at the transcriptomic level. This allows it to thrive ecologically in both high-salinity and low-salinity environments.

Last but not least, we have outlined a list of fundamental genes related to salinity that offer potential for enhancing tolerance to saline conditions. The array of genes unveiled in this study has shown significant effectiveness in enhancing plant tolerance to salt stress by manipulating a single gene, offering hope in addressing concerns related to genetically modified organisms by creating organisms through genome editing that are indistinguishable from conventionally bred varieties. Unfortunately, only two genes have been confirmed in both microalgae and higher plants, indicating the need for further exploration. Despite this limitation, this research provides essential genetic reservoirs crucial for the advancement of halotolerant plants.

## Limitations of the study

We acknowledge the intricate and multifaceted nature of habitat transitions in Viridiplantae (including green algae), which encompass not only shifts from marine to freshwater environments but also independent movements from freshwater or terrestrial habitats to marine settings. While numerous lineages of Viridiplantae encompass both marine and freshwater species, our choice of *Chlorella* as a model for examining transition mechanisms stems from its remarkable ecological diversity and phylogenetic significance. However, we are unable to definitively establish a direct correlation between these adaptive characteristics and the transition from saltwater to freshwater environments. Nonetheless, it is highly probable that these specific adaptations have played a facilitating role in the transitions between marine and freshwater conditions.

## Methods

### Growth conditions

*Chlorella* sp. strain MEM25 was maintained in dim light at 4 °C on solid modified F2 medium, which was prepared as the early report[40]. *C. pyrenoidosa* FACHB-9 was maintained in BG11 medium (1500 mg L$^{-1}$ NaNO$_3$, 30 mg L$^{-1}$ K$_2$HPO$_4$, 75 mg L$^{-1}$ MgSO$_4$·7H$_2$O, 36 mg L$^{-1}$ CaCl$_2$·2H$_2$O, 6 mg L$^{-1}$ ferric citrate, 6 mg L$^{-1}$ ammonium citrate, 1 mg L$^{-1}$ EDTA, 20 mg L$^{-1}$ Na$_2$CO$_3$, 2.86 mg L$^{-1}$ H$_3$BO$_3$, 1.81 mg L$^{-1}$ MnCl$_2$·4H$_2$O, 0.222 mg L$^{-1}$ ZnSO$_4$·7H$_2$O, 0.390 mg L$^{-1}$ NaMoO$_4$·5H$_2$O, 0.079 mg L$^{-1}$ CuSO$_4$·5H$_2$O, 0.0494 mg L$^{-1}$ Co(NO$_3$)$_2$·6H$_2$O)[22], at 25 °C under a 12-h light/12-h dark photoperiod (50 μmol·photons·m$^{-2}$·s$^{-1}$). For inoculation, cells were transferred into the liquid medium and were maintained under illumination of 50 μmol photons m$^{-2}$ s$^{-1}$ at 25 °C with a light/dark cycle (12 h/12 h).

### Chromosome staining

Algal cells in mid-logarithmic phase were collected and treated with 0.1% w/v colchicine for 6 hours, followed by fixation in Carnoy's solution for 12 h. Thereafter, the cells were washed in 70% methanol and fresh F2 medium. They were then treated with a softening solution containing 5% EDTA with 0.1% Tween 20 for 1 h, followed by hydrolysis with 30 mM KOH and 500 mM HCl at 50 °C for 1 h, respectively. The cells were collected and suspended in 0.1 M citrate buffer (pH=5) with Cellic®Htec$^2$ (Novoenzymes) at 37 °C for 1 h. Next, the cells were immediately stained with 0.5 mg mL$^{-1}$ 4′, 6-diamino-2-phenylindole (DAPI) for 5 min and were examined under a microscope[41].

### Genome assembly

Single molecule real time sequencing (SMRT) libraries and Illumina short read libraries were constructed using fresh mid-log phase algae (BioMarker, China), followed by sequencing with the PacBio Sequel II and HiSeq 2000 platforms. The Hi-C library and sequencing were conducted with the Illumina HiSeq platform (Novogene, China). Multiple approaches have been employed to guarantee the quality of genome sequences. PacBio and Hi-C data, Hifiasm (version 0.16.1-r375) was used for subgenome splitting. SMRT sequencing data were assembled using Falcon (version 0.3.0) and CANU (version 2.1.1). 3D-DNA (version 180922) and ALLHiC (version 0.9.8) were used for draft genome correction and genome assembly, respectively. Finally, Juicebox (version 1.11.08) and Pilon (version 1.24) were used to correct genome fragments (based on the chromosome contact frequency matrix) and base errors of the genome with Illumina and RNA-seq data, respectively. The Hi-C interaction frequency matrix was analyzed by Homer, FitHic (version 2.0.8), HiCExplorer (version 3.7.2), and HiC-CUPS (version 5.14). The 3-D chromosome coordinate location was calculated using HiC-GNN and was visualized using UCSF Chimera (version 1.15).

### Genome annotation and analyses

For genome annotation, various references were used, including *A. thaliana*, *Auxenochlorella protothecoides* UTEX25, *Chlamydomonas reinhardtii*, *Chlorella sorokiniana* 1602, *Chlorella variabilis* NC64A, *Coccomyxa subellipsoidea* C-169, *Dunaliella salina* CCAP1918, and *Micractinium conductrix* SAG241. Augustus (version 3.3.3), GlimmerHMM (version 3.0.4), SNAP (version 2013.11.29), Geneid (version 1.4), and Genscan were used to predict gene structure ab initio. Cufflinks (version 2.2.1) was used to predict novel genes and correct gene structure. EVidenceModeler (version 1.1.1) and PASA (version 2.4.1) were used to remove redundancy and correct the integrated annotation results.

Gene function was inferred using BLASTP (with a cutoff of *e* value < 1 × 10$^{-5}$) by comparing with known protein sequence databases, such as Evolutionary genealogy of genes: Non-supervised Orthologous Groups (eggNOG), National Center for Bio-technology Information (NCBI), International Protein Resource Information System

(InterPro)[42], The Protein Families Database (Pfam), SwissPro, and KEGG.

## Conserved non-coding DNA sequence annotation

The repeat sequences were annotated using RepeatMasker (version 4.1.0) while conserved non-coding DNA sequences (CNSs) were identified using dCNS (version 0.4). For CNSs annotation, the reference genomes included those of *Chlorella* sp. A99, *A. thaliana*, *A. protothecoides* UTEX-25, *C. reinhardtii*, *D. salina*, *C. sorokiniana* 1230, *C. pyrenoidosa* FACHB-9, *M. conductrix* SAG241, *C. variabilis* NC64A, *Ostreococcus tauri*, and *Picochlorum soloecismus* DOE101. The short segment tandem repeat sequences were eliminated and the adjacent CNSs with regional overlap or with spans less than 50 bp were merged. Finally, a total of 721 CNSs was obtained.

## Telomere and centromere scanning

Tandem repeats finder (TRF, version 4.09) was used to search for tandem repeat sequences and identify telomere sequences. For centromere annotation, minimap2 (version 2.2.26) was used to mask the functional regions, followed by removal of redundancy and searching and extraction of the repeated units. The genome sequences were segmented into fragments with a length of 500 bp. CD-hit (version 4.8.1) was used to find the sequences with similarities above 70%. MEME SUITE (version 5.3.0) and Homer were used to extract the featured fragments of these sequences, which were subsequently aligned to the individual chromosome. The sequence similarity was visualized by Circoletto.

## Genome quality assessment and collinearity analysis

LTR_FINDER (version 1.07) was used to identify Long terminal repeat retrotransposons (LTR-RT), this comprises a non-redundant LTR-RT library, with LTR Assembly Index (LAI) values calculated. Gene annotation and assembly were evaluated via BUSCO (version 5.2.2) using the single copy Homeotic gene database of Chlorophyta (https://busco-data.ezlab.org/v5/data/lineages/chlorophyta_odb10.2020-08-05.tar.gz). The genome synteny and collinearity were detected by JCVI (version 1.3.5) and the results were visualized by the R package Rideogram (version 0.2.2).

## Comparative genomic and phylogenetic analysis

Genome sequences for the selected species were retrieved from the NCBI and IMG-M databases (https://img.jgi.doe.gov/cgi-bin/m/main.cgi) (Supplementary Data 4). OrthoFinder2 (version 2.5.2) was used for orthogroup inference, with 39,321 orthogroups obtained for the 46 selected species. Among them, 199 orthogroups were present in all the selected species, and these were used to infer species trees by using the STAG algorithm. Dollo parsimony was implemented for Count to infer family- and lineage-specific characteristics across the evolutionary tree.

Chloroplast genome sequences were retrieved from the CGIR database (https://ngdc.cncb.ac.cn/cgir/). Parsed hits for all species were aligned using MAFFT7 (version 7.480). Gaps and ambiguously aligned sites were removed using gBlock (version 0.91b). Sequences that caused aberrant alignments and whose real identity could not be confirmed were removed manually. Phylogenetic analyses were performed with a maximum likelihood method using IQ-TREE (version 2.1.4-beta).

The R8S script was employed to estimate temporal divergence based on the molecular evolution rate and stable fossil nodes. A strict clock model was used to avoid horizontal gene transfer and other events that affected divergence times. Multiple time constraints (fossil records for no less than three species within the same genus) were incorporated to evaluate our results using fossil cross-validation. The fossil records used in this study are relevant ones that have previously been applied to estimate the divergence times of eukaryotes, including those for (1) *A. thaliana* to *Oryza sativa* (115-308 Mya), (2) *Mesostigma viride* to *Chlorokybus atmophyticus* (174-631 Mya), (3) *Ostreococcus lucimarinus* to *Micromonas pusilla* (333-639 Mya), (4) *Synechocystis* sp. PCC 6803 to *Synechococcus moorigangaii* (1580-2557 Mya), and (5) *Porphyra umbilicalis* to *Cyanophora paradoxa* CCMP329 (1386–1680 Mya). The fossil records are available at Timetree (http://www.timetree.org/).

## Genome-wide association

A mutant pool for MEM25 was generated by EMS mutagenesis following our previous report. In brief, the mutagenesis conditions were optimized by treating log-phase cells with different concentrations of EMS for varying durations, followed by assessing the number and morphology of the resulting colonies. A mutant pool was constructed using the optimized conditions and candidate mutants were further assessed for growth in either the low salinity (i.e., 35‰) or the high salinity (i.e., 70‰) conditions. The turbidity was monitored by recording $OD_{750}$ at indicated intervals. A total of 536 mutants with altered $OD_{750}$ were selected for further phenotyping under the high-salinity conditions. The 365 mutants with stable phenotypes after ten generations were assessed further for growth in the high salinity conditions (i.e., 70‰). Dry weight, cell size, and cell number were recorded. Finally, genomes of 195 salinity-related mutants and three WT samples were resequenced using the second-generation sequencing Illumina platform for PE150 sequencing (Novogene Company, China).

Fastp (version 0.20.0) was used for filtration and quality control of original sequencing data (fastp -t 10 -f 10 -F 10 -T 10 -W6 -u 20 -n 10 -c). Genome alignment was conducted using BWA-MEME (version 0.7.17-r1188) and SAMtools (version 1.12) (bwa mem -R '@RG\tID:{sample} \tSM:{sample}\tPL:Illumina'\ index fq1 fq2 | samtools sort -@ 2 -m 1 G | samtools view -h -b -q30 > q30.sort.bam), followed by quality control using Qualimap (version 2.2.2). PCR duplicates and optical duplicates were removed using Picard's MarkDuplicates program (https://broadinstitute.github.io/picard/). Only high-quality sequencing data were used for the subsequent analysis. Variations were detected using Genome Analysis Toolkit (GATK) (version 4.2.1). A value of 0.00743 was obtained for the genomic heterozygosity of MEM25 by using a combination of Jellyfish (version 2.3.0) and GenomeScope (version 2.0). Genome alignment on the sequences around INDEL (Insertion and Deletion) was performed to detect variations for each sample. The results of all samples were merged when population variations, SNPs, and INDEL were obtained. Quality control on SNPs (QUAL < 30 || MQ < 40.00 || SOR > 4.000 || QD < 2.00 || FS > 60.000 || MQRankSum < −10.000 || ReadPosRankSum < −10.000 || ReadPosRankSum > 10.000) and INDEL (QUAL < 30 || MQ < 40.00 || SOR > 10.000 || QD < 2.00 || FS > 200.000 || ReadPosRankSum < -20.000 || ReadPosRankSum > 20.000) was conducted separately. Variations with abnormal sequencing depth were removed. GWAS analysis was performed using TASSEL, employing a Mixed linear model and Bonferroni correction. Variations were annotated via SnpEff. R package CMplot (https://github.com/g-insana/CMPlot.jl) was used to draw a Manhattan plot, Quality-Quality plot, and genome variation distribution maps.

## Sample preparation for transcriptome and metabolome analysis

Cells were cultured into log phase under optimal conditions. Salt stress was induced by transferring microalgae to high-salinity conditions (that is 35 g·L$^{-1}$ for FACHB-9 and 105 g·L$^{-1}$ for MEM25). Aliquots of cells were collected following the salinity shifts for either transcriptome (after 3 h and 24 h) or metabolome (after 24 h) analysis. Triplicates were used for each treatment.

## Transcriptome analysis

Total RNA extraction, sequencing, and assembly were performed according to previous reports[40]. Transcriptome sequencing was

completed by BioMarker Company (China, Beijing). Quality control and removal of low-quality fragments was conducted by FastQC (version 53.0) and fastp, followed by the trimming of low-quality regions via Trimmomatic (version 0.39). The RNAseq data were compared with the reference genomes using GSNAP (version 2021-07-23), followed by quantification of the values of FPKM (Fragments Per Kilobase of exon model per Million mapped fragments) using StringTie (version 2.1.7), R package Ballgown (version 2.22.0), and DESeq2. Differentially expressed genes were defined as those with FDR-adjusted $p$-value $\leq 0.05$ and $|\log_2 FoldChange| \geq 1$. Function and pathway enrichment analysis was conducted using KEGG and the R package clusterProfiler (version 3.18.1).

### Metabolome analysis

~500 mg of algal biomass was collected, followed by metabolite extraction. Algal cells were suspended in 1 mL mixture of methanol and water (7/3, vol/vol) and kept at $-80\,°C$ for 2 min. The internal standard used was 2-chloro-l-phenylalanine ($0.3\,mg\,mL^{-1}$). The mixture was vortexed at 60 Hz for 2 min, followed by ultrasonication at ambient temperature for 30 min. Samples were then centrifuged at 13,000 g, 4 °C for 15 min. The supernatants were collected using crystal syringes, filtered through 0.22 μm microfilters and transferred to LC vials, followed by storage at $-80\,°C$. LC–MS/MS analyses were performed using an UHPLC system (1290, Agilent Technologies).

### Metabolite-transcript correlation analysis

Pearson correlation coefficients (PCCs) were calculated for metabolite and transcript profiles as described previously[43]. The mean of all the biological replicates for individual metabolites and the normalized mean value of the transcriptional levels of each gene were evaluated. The coefficients were calculated using the $\log_2$ (fold change) values, with PCC > 0.90 and PCC $P$ value (PCCP) < 0.001 used as the cutoff. The connection network between orthogroups and metabolites was built via Cytoscape (version 3.8.2).

### Weighted gene co-expression network analysis

Each orthogroup, out of the 5026 shared by MEM25 and FACHB-9 (including 3101 single copy orthogroups), was designated as a metagene. Taking into account both the transcriptional level and the gene copy number's impact, the metagenes' transcriptional levels were quantified by summing the transcriptional levels of all genes within each orthogroup. The WGCNA version 1.70-3 was used to analyze the transcriptome and metabolome data, with the orthogroups categorized into different modules, correlated with specific trait(s). A cutoff was set with the correlation coefficient > 0.75 and a $P$ value < 0.005. For the transcriptome, the WGCNA soft threshold of the expression matrix was set at 7, with the scale-free topological model fitting $R^2 > 0.9$. Regarding the metabolome, the values were 9 and 0.9, respectively.

### Principal component analysis

Principal component analysis of featured genes was performed using the R package ggord (https://zenodo.org/badge/latestdoi/35334615). The packages prcomp and factoextra were used for the analysis and ggplot2-based visualization. The abscissa of the PCA score chart represents the first principal component, namely PC1, and the ordinate represents the second principal component, namely PC2. The confidence interval level for PCA analysis was set to 0.95.

### Machine learning analysis

A machine learning analysis was conducted on a gene family presence-absence matrix across the 46 species to discern crucial genetic traits that differentiate freshwater and marine algae. Each species was represented in binary form, with "1" denoting the presence of a gene family and "0" indicating its absence. Our method involved a two-stage machine learning approach encompassing 86 models and model combinations, including Random Forest, Lasso, glmboost, Ridge, SVM, KNN, NaiveBayes, GBM, DecisionTree, and Enet (with various alpha values). Initially, models performed feature selection to pinpoint potential gene families, followed by additional filtering or classification to enhance classification accuracy. A 5-fold cross-validation method was employed across all models to ensure robust performance evaluation. For models like Ridge Regression without inherent feature constraints, accuracy was incrementally tested with feature sets in batches of 50 genes to determine the optimal feature count for peak performance. This meticulous approach ensured that each model combination achieved its highest classification accuracy, enabling precise identification of genetic traits crucial for distinguishing between freshwater and marine species. The most effective model combination, Random Forest + Elastic Net [0.6], boasting a classification accuracy of 97%, was utilized to highlight featured genes.

### Cas9-mediated gene knockout and phenotype assays

Cas9-mediated target gene disruption in *Nannochloropsis* was conducted using an episomal CRISPR system[44] following a protocol described previously[40]. Two sgRNAs were designed for each gene. For each transformation reaction, $10^9$ log-phase cells were mixed with 1 μg cassette in an electroporation cuvette (2 mm gap). Electroporation was performed using the GenePulse Xcell™ (Bio-Rad) with $11\,kV\,cm^{-1}$ field strength. After the pulse, the cells were immediately mixed with 5 ml F2 medium for recovery. Then, the cells were plated on the F2 agar plate with $2\,μg·mL^{-1}$ zeocin until transformant colonies appeared. Mutations with target disruption were screened by PCR with specific primers (Supplementary Data 20). The growth of microalgae was monitored by measuring the turbidity or cell number at specified intervals with a Gene Quant 1300 Spectrophotometer (GE) or LUNA-II™ Automated Cell Counter (Logos Biosystems). Dry cell weight of a 10-ml algal culture was determined simultaneously.

### *A. thaliana* mutants and phenotype

*A. thaliana* Col-0 (WT) and SALK_202085C (*cnbp*) T-DNA insertion mutant lines were obtained from The Nottingham *Arabidopsis* Stock Center (NASC). The T-DNA insertion position and homozygous lines were verified according to instructions from the *Arabidopsis* Biological Resource Center (https://abrc.osu.edu/help/genotyping). The genomic sequence of the mutated gene was used to design the forward primer while the T-DNA left border sequence was used to design the reverse primer (LBa1, 5'-tggttcacgtagtgggccatcg-3'). Seeds were surface sterilized and germinated on half-strength MS medium containing 3% sucrose and 0.8% phytoagar (pH 5.7) in a growth chamber at 22 °C with 100 μmol·photons·$m^{-2}·s^{-1}$ light and a 16 h light/8 h dark cycle. Two-week-old seedlings were used for NaCl treatments (150 mM). Seeds of the wild-type and mutant plants were grown on the same plates containing MS medium with or without NaCl. Plants were grown at 22 °C with 100 μmol·photons·$m^{-2}·s^{-1}$ light and a 16 h light/8 h dark cycle. The root length was measured with a ruler at the specified times.

### Reporting summary

Further information on research design is available in the Nature Portfolio Reporting Summary linked to this article.

## Data availability

All data supporting the findings of this study are publicly available. The Whole Genome Shotgun project for MEM25 has been deposited at NCBI GenBank under the accession JBSQDC000000000, which includes the complete genome sequence. Raw sequencing data are available under NCBI BioProject accessions PRJNA1344311 (sequencing data for assembly, including Illumina, Hi-C, and PacBio data) and PRJNA1344302 (GWAS mutant resequencing data comprising 198 samples). The RNA-seq data have been deposited under the NCBI BioProject accession PRJNA1370502, including raw transcriptome

sequencing data from a total of 24 samples of MEM25 and FACHB-9. The metabolomics data are deposited in the EMBL-EBI MetaboLights database under accession MTBLS13425, which encompasses raw metabolomics sequencing data, processed metabolite contents, and corresponding annotations derived from a total of 12 samples of MEM25 and FACHB-9. The sequences included in the comparative genomic and phylogenetic analysis of this study have been made available via FigShare with (https://doi.org/10.6084/m9.figshare.30811226) (https://doi.org/10.6084/m9.figshare.30811226). Additionally, the integrated multi-omics data of MEM25 used in this study have been deposited in Figshare with (https://doi.org/10.6084/m9.figshare.29062721) (https://doi.org/10.6084/m9.figshare.29062721). Source are provided with this paper. Source data are provided with this paper.

## Code availability

The bioinformatics code for data analysis is archived in Zenodo (https://doi.org/10.5281/zenodo.17388984), and all source data and code required to regenerate the figures are available on GitHub (https://github.com/aoqiwang/MEM25).

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

## Acknowledgements

We thank Michael Melkonian (Department of Plant Microbe Interactions, Max Planck Institute for Plant Breeding Research, Cologne, Germany) and Sibo Wang, Linzhou Li, and Yan Xu (BGI Genomics, China) for their valuable contributions to data analysis and for enhancing the manuscript. We thank Jianhua Fan (East China University of Science and Technology) for sharing the genome data of *Chlorella pyrenoidosa* FACHB-9. **Funding** This work was supported in part by grants from the National Key R&D Program of China (2021YFA0909600), the National Natural Science Foundation of China (32560020 and 32370380), the Key R&D Program of Hainan Province (ZDYF2024XDNY244), the Natural Science Foundation of Hainan Province (322QN250), the Foreign Expert Foundation of Hainan Province (G20230607016E), and the Hainan Tropical Ocean University Joint Open Project for Aquatic South Breeding (2023SCNFKF04).

## Author contributions

Y. L. conceived and planned the experiments. Y. L. took the lead in writing the manuscript. Q. W. contributed to sample preparation and bioinformatics analysis. Q.G. contributed in sample preparation and performing the experiments related to *Arabidopsis* phenotyping. Y. X. contributed in performing the experiments related to *Nannochloropsis* mutant creation. Y. D. contributed in performing the experiments related to *Nannochloropsis* mutant phenotyping. X. H. contributed in sample preparation. All authors provided critical feedback and helped shape the research, the analysis, and the manuscript.

## Competing interests

The authors declare no competing interest.
