## [Transparent Peer Review file · Nature Communications]

Cross-species dissection of saline-related genes by genetically deciphering a euryhaline microalga *Chlorella* sp.

Corresponding Author: Dr Yandu Lu

Version 0:

Reviewer comments:

Reviewer #1

(Remarks to the Author)

The manuscript titled "Cross-species dissection of saline-genes by genetically deciphering a euryhaline microalga" demonstrates promising research but requires several major revisions. First, the genus name should be incorporated into the title. While the study effectively compares multiple species, it lacks detailed analysis of saline-genes and their evolutionary patterns across species. The introduction needs strengthening, particularly regarding the rationale for selecting *Chlorella* genome and its ecological significance in freshwater/seawater environments.

The manuscript's structure and organization require substantial modification. The supplementary materials are excessive (27 supplementary Tables), making it difficult for readers to navigate. I recommend consolidating tables and utilizing public depositories for supplementary data. The manuscript currently exceeds Nature Communication's guidelines, containing 11 main figures (exceeding the 10-figure limit) and 118 references (surpassing the 70-reference maximum), with methodology citations comprising more than half of the references.

The main text is overly lengthy at 30 pages and exceeds 5,000 words, with the Results & Discussion section being particularly verbose. The authors should condense the content while maintaining the scientific integrity of their findings. Despite these structural issues, the study presents interesting findings and valuable results. However, addressing these formatting and organization concerns would significantly improve the manuscript's clarity and adherence to journal guidelines.

The authors should address the following points for clarification:

- L.3: Clarify the meaning and reference of the # symbol
- L.49: Add missing "Introduction" header
- L.59-60: Rephrase sentence to specify which "other eukaryotes" are being discussed
- L.74-75: Content needs clarification/completion
- L.88-97: Include methodology introduction for addressing stated hypotheses

General comments: Introduction is too brief compared to Results & Discussion and Discussion sections. Please add justification for microalga species selection and expand methodology overview for addressing research hypotheses

Results and discussion

- L.110: Delete "s" from "genomes".
- L.113: Specify databases used.
- L.116-117: The transcriptome has not been presented before, nor mentioned in the first paragraph of the results. How was the transcriptome done?
- L.122: Define T2T acronym.
- L.140-142: Add references for known species genomes mentioned in the text.
- L.151-152: How have the authors performed the whole chromosome interactions, and should they briefly mention the methods used here, e.g. correlation using which metrics or algorithm?
- L.152: Remove or justify "As expected" statement.

- L.154-156: Include literature citations.
- L.165-168: Add citations for species selection.
- L.169: Specify and cite Cyanobacteria species used in the analysis.
- L.175: Fix spacing between "outgroups" and "(Table S10)". Concerning Tables S10: The authors should consider transforming their Table S10 into a main table so that the reader can directly see which species they have chosen for comparison, as the authors have not included all the names of the species used in the main text.
- L.183: Reference table/figure for 13 ortholog genes.
- L.186: Add missing Streptophyta color in Fig.3a.
- L.197: Replace "exceptional position" with more appropriate wording.
- L.227-228: Include method reference.
- L.261-262: Specify relevant table/figure.
- L.273: Introduce neutrophilic freshwater FACHB-9 and cite the literature.
- L.277: Specify *C. reinhardtii* strain name.
- L.318-320: Where can we see that these genes are salt-sensitive and not associated with cell damage? This is not clear.
- L.327-328: Same question as above, provide data location.
- L.435: Correct table citation order (S24 before S25).
- L.548: Reorganize Figure 11 panels for better readability.

The Results and Discussion section's first part needs more thorough analysis to effectively convey the findings' implications.

Discussion

There is inconsistency in the manuscript's structure regarding Results and Discussion sections. At L.98, the section is labeled as "Results & Discussion," yet at L.549, a new "Discussion" section appears. Authors should choose either a combined Results & Discussion format or separate Results and Discussion sections. Given the limited discussion within the Results section, separating these sections would be more effective. This would allow for a focused presentation of findings in the Results section, followed by a comprehensive Discussion section that fully develops the interpretation and implications of all results. Throughout the Discussion section, the authors do not mention any figures or tables to remind the reader where to find the information discussed! Please correct this.

- L.553: Remove "exceptional" unless providing comparative literature support for this characterization.
- L.553-555: Add specific table references for coverage information.
- L.558: Define LAI acronym at first use.
- L.608: Move FPKM definition to its first appearance, which should be in the Results section.
- L.621-623: Move methodological information to Methods and/or Results sections.
- L.661-662: Define acronyms GPHGs and Ens at first mention.
- L.722: Authors have mentioned this gene before, but have not written it in italics, except in L.722, 725, but then in L.727 the gene name is no longer in italics. Please correct this.

Conclusion

The conclusion section requires substantial revision as it fails to adequately summarize the study's key findings and does not address the two main hypotheses presented in the introduction. A more comprehensive conclusion should be developed that explicitly connects the research outcomes to the initial hypotheses and synthesizes the major discoveries. This expanded conclusion would provide readers with a clearer understanding of how the study's findings contribute to the field and fulfill its original objectives.

Reference

The manuscript exhibits significant citation imbalances, with only 49 out of 118 references supporting the core content across introduction, results, and discussion sections. The discussion section is particularly under referenced, utilizing just over 10 citations (references 38-49), indicating insufficient engagement with existing literature and inadequate result analysis. While the methodology section is extensively referenced, it needs condensing.

Reference formatting requires standardization throughout, as there are inconsistencies in DOI presentation, journal name abbreviations, and please check Nature Communication citation styles.

Data availability

Data accessibility needs improvement through the provision of complete accession numbers and relevant links. If custom bioinformatics code was used, it should be deposited in a public repository.

Regarding figures, the caption for Figure 11 exceeds Nature Communications' 350-word limit and requires condensing.

The legend at L.1658 incorrectly uses identical symbols for filled circles, necessitating correction.

Reviewer #2

(Remarks to the Author)

This Manuscript describes the mechanisms and acquired process of the euryhaline in an undescribed species of *Chrorella* that can grow in a wide range of salinity environments. The manuscript presents an impressive volume and variety of data,

with extensive analyses that are truly remarkable. I noticed that several expressions are uncommon in the context of scientific writing and molecular evolution, which affects the clarity of the text. Below, I outline my specific points of interest.

Major concerns

1.
The T2T genome of MEM25 looks substantially accurate and complete. I have some concerns about its statistics. Although they are minor mistakes, they might not be as reliable as they should be.

There are inconsistent descriptions about N50 and contig length.

- scaffold N50 as 4.3 Mb in MS line 107
- the longest as 4.1 Mb in MS line 107
- contig N50 as 3.76 Mb in Table S1
- contig N50 as 4.1 Mb in table S2

There is no information on the longest contigs in Table S2.

N50 is never longer than the longest contig (or scaffold) length because N50 is defined as the sequence length of the shortest contig at 50% of the total assembly length.

Please check the inconsistency of these statistics and reanalyze related calculations if necessary.

I can find the BUSCO score with genome mode on line 115 and Table S5.

Did you calculate the BUSCO score with protein (or transcriptome) mode?

Although I will not see a problem since the BUSCO score with the genome is high, I would like to confirm it for the following analyses based on the gene prediction.

There is no mention of gap regions despite scaffolding with Hi-C. Is the T2T genome assembled completely gapless? If so, please appeal the completeness with a sense of pride.

2.

You found that gene families were expanded in MEM25. Would MEM25's assembly completeness and high-quality gene prediction overestimate the number of genes?

I agree with the analysis based on the presence-absence matrix of gene families. However, the technological limitations of previous studies and the difficulty of assembling the algae genome might influence the results of the gene family expansion analysis.

If there are serious differences in the quality of gene prediction among species, differences in FPKM caused by the mapping accuracy of RNA-seq may be observed. Couldn't the false correlation that larger gene families have a higher expression level affect the co-expression analysis in this study?

3.

The centromere in land plants is defined by CENH3 binding. It is confirmed with ChIP-seq because the centromere is located at multiple positions and in different repeat patterns and positions among taxons.

I'm unsure about centromere systems in Chlorophyta. Is it truly centromere? It looks like kinds of "centromere-like tandem repeat."

4.

I understand that MEM25 has habitat-related genes in both saltwater and freshwater environments. I'm not sure that "MEM25 represents an early-diverging saltwater species".

I agree with the MEM25's earlier divergence of the MEM25 in the genus *Chlorella*. However, I wonder if it is early in this study's huge variety of algae. It remains unclear whether this phylogenetic relationship indicates that the MEM25's saltwater habitat is derived from ancestral traits or was reacquired. Based on your scope, I would like you to clarify the point of "early-diverged", focused on the timing comparing what and the directory.

Minor concerns

Line 139-149

The absence of WGD and sequence conservation of centromeres and telomeres is unclear from Fig 2c. Almost all Angiosperm, including *A. thaliana* have experienced WGD in the late Cretaceous. If the correspondence with *A. thaliana* are found, MEM25 (and other *Chlorella* species) could have been duplicated as well.

And the telomeres of MEM25 seem to have the synteny to only one region in FACHB.

Line 384-388

I could not find these five families in Table S21. And OG0012359 was mentioned twice as a different term. Please confirm the gene family classification.

Line 392

You mention “orphan genes” while you mention the gene function. Which is a true orphan genes or orthologous genes?

Line 466

What does “DW” mean?

Line 556

A N50 value depends on the genome and chromosome size. A relative comparison is valuable, but an absolute value makes no sense.

Line 560

The chromosome-level assembly lacks several telomere repeats, right? I guess that it is difficult to find all telomere repeats in this case, especially telocentric chromosomes like chr15.

Line 760-765

I agree with the idea that MEM25 has a mechanism different from *D. salina*. As a possibility, euryhalinity (not halophilism) might be explained by sodium excretion rather than salt tolerance.

For example, I know a case of adaptation to a marine beach in *Vigna marina*. I don't think the same mechanism from the same origin because it is embryophytes, but this mechanism could protect from salt and grow a wide range of salt concentrations.

Diurnal Regulation of SOS Pathway and Sodium Excretion Underlying Salinity Tolerance of *Vigna marina*

Noda et al. (2025)

<https://doi.org/10.1111/pce.15402>

Section Genome-wide association in line 930

Does the mutant pool share SNPs? Do multiple individuals obtain mutations generated by EMS mutagenesis in the same positions?

Please let me know if this is a sample set that can be used for GWAS.

line 946

Does BWA-MEME mean BWA-MEM?

line 982-983

You show the thresholds of P-adj and FDR. What does P-adj mean, not FDR?

line 945

The parameters of fastp “-t 10 -f 10 -T 10 -F 10” appear to be over-trimmed. Is there any reason?

line 958

Did you use these QC options? or filter out them?

Figure 3a

Do these legend colors indicate the background of the species' names? Please use more distinctive colors.

Figure 3b

The positions of the purple circles are unclear, and some seem to locate no branch points.

Figure 6b

Green indicates bacteria. However, other colors also indicate the kind of bacteria.

Figure 9

Please add the details. For example, what are the orange points in (d)? Where is the gene in (e)?

Figure S4a shows the polar tree rather than the unrooted tree. Polar, the center of the figure, generally indicates the root.

Legend Fig. S5

line 1658

What do the “filled circles ● and ●, respectively” mean? It looks like the same filled circles to me.

Reviewer #3

(Remarks to the Author)

Overall, this is a comprehensive and well written manuscript. A high-quality chlorella genome will be useful to the community. Despite extensive data assembly and analysis, the manuscript could be improved. Specifically, when increased biomass is reported this should be further quantified in terms of lipids, carbohydrates or proteins (or more cells) to better understand how more biomass is being made. Additionally, more information regarding RMI1 is encouraged. A thorough protein map with putative domains is necessary, as is alignments with homologs. Lastly, a more thorough hypothesis regarding RMI1 mechanism is encouraged.

Version 1:

Reviewer comments:

Reviewer #1

(Remarks to the Author)

The authors of the manuscript entitled "Cross-species dissection of saline-related genes by genetically deciphering a euryhaline microalga *Chlorella* sp." have submitted a detailed revised manuscript addressing the reviewers' comments. I appreciate the extensive effort they have invested in this revision process. However, several modifications are still required before publication:

The methodology section requires significant improvement. Currently, it merely lists section headings without providing adequate experimental details. Authors must develop each methodological section with sufficient information to ensure reproducibility. If the comprehensive methodology would make the main text too lengthy, detailed protocols should be provided as supplementary files. The current format, which presents only section titles without substantive content, is inappropriate for a scientific publication. Please revise this section to include complete methodological descriptions.

Reviewer #2

(Remarks to the Author)

I am highly impressed with the completeness of the T2T-level assembly, notably achieving a single gap. Is there any support for a gap length of 100 bp? The software might arbitrarily insert the round number. Although the quality of the genome with a single gap is not compromised, if a 100bp gap is an artifact, it is appropriate not to mention it.

Fig. 2

Distinctive colors make this figure clear. However, the background colors are in the front, making the characters hard to see. Please exchange the layers' orders.

Reviewer #3

(Remarks to the Author)

The authors have adequately addressed my concerns.

Version 2:

Reviewer comments:

Reviewer #1

(Remarks to the Author)

I would like to commend the authors for addressing all of the editor's and reviewers' comments thoroughly. The improvements to the manuscript are clear, and the authors have done an excellent job of revising it.

**State Key Lab of Marine Resource
Utilization in South China Sea, Hainan**

**Mailing: 58 Renmin Avenue
Haikou, Hainan, China, 570228**

**Tel: 86-898-66257612
Fax: 86-898-66251258**

April 9, 2025

Dear Reviewers,

Thank you for the constructive and extremely valuable comments to help improving the manuscript. Here we are submitting a revised version, with item-by-item responses (**bold fonts**) to the comments below.

We have thoroughly proofread and edited the text following the requirements. We believe that our revised version supports the main conclusion of our initial submission and also reveal additional insights, which were detailed below.

Comments from Reviewer 1:

1. The manuscript titled “Cross-species dissection of saline-genes by genetically deciphering a euryhaline microalga” demonstrates promising research but requires several major revisions. First, the genus name should be incorporated into the title. While the study effectively compares multiple species, it lacks detailed analysis of saline-genes and their evolutionary patterns across species. The introduction needs strengthening, particularly regarding the rationale for selecting *Chlorella* genome and its ecological significance in freshwater/seawater environments.

The manuscript’s structure and organization require substantial modification. The supplementary materials are excessive (27 supplementary Tables), making it difficult for readers to navigate. I recommend consolidating tables and utilizing public depositories for supplementary data. The manuscript currently exceeds Nature Communication's guidelines, containing 11 main figures (exceeding the 10-figure limit) and 118 references (surpassing the 70-reference maximum), with methodology citations comprising more than half of the references. The main text is overly lengthy at 30 pages and exceeds 5,000 words, with the Results & Discussion section being particularly verbose.

The authors should condense the content while maintaining the scientific integrity of their findings. Despite these structural issues, the study presents interesting findings and valuable results. However, addressing these formatting and organization concerns would significantly improve the manuscript's clarity and adherence to journal guidelines.

Response: We greatly appreciate the reviewer’s inputs and we are sure that these inputs greatly help us improve the manuscript. We have carefully revised as advised to improve the manuscript's clarity and adherence to journal guidelines.

State Key Lab of Marine Resource Utilization in South China Sea, Hainan

Mailing: 58 Renmin Avenue
Haikou, Hainan, China, 570228

Tel: 86-898-66257612
Fax: 86-898-66251258

We have removed or reworded the Results, Discussion and Methods sections when necessary.

- Title:

As advised, we have incorporated the genus name:

Previous title: Cross-species dissection of saline-related genes by genetically deciphering a euryhaline microalga

Current title: Cross-species dissection of saline-related genes by genetically deciphering a euryhaline microalga *Chlorella* sp.

- Introduction:

We have strengthened the content regarding the rationale for selecting *Chlorella* genome and its ecological significance in freshwater/seawater environments. Now the text reads (Line 67 to line 83 in current version):

“To shed light on the genetic and cellular adaptations that have allowed crossing over the salinity barrier, the generation of genomic data from closely related marine and freshwater lineages is crucial ¹. An appropriate choice of organism group is essential for such studies. Marine-type sister groups of Streptophyta have not been confirmed ². Therefore, it is essential to find a speciose group with ample genome data available, occurring in both freshwater and saline water.

Chlorophyta microalgae of the genus *Chlorella* seem to satisfy these criteria. Their minute cell size and resistance to environmental stress facilitate long-distance dispersal. The genus is omnipresent in terrestrial and aquatic habitats ³. Although the majority inhabit freshwater ⁴, a large number of species occur in the oceans, from the Antarctic, to temperate regions and the tropics ⁵. However, the genomes of saltwater *Chlorella* spp. have not previously been sequenced. The euryhaline *Chlorella* sp. MEM25 (hereafter MEM25) was originally isolated from the inland saline water of Hainan Island, China, in August 2016 ⁶. The water is highly saline (> 65‰), with a year-round high temperature (from 30 to 41 °C). MEM25 grows vigorously across a broad range of environmental conditions, including in salinities ranging from 0 to 105‰. With increasing salinity, maximum biomass production was observed at 70‰ ⁶.”

Moreover, we also acknowledge the limitations of the study in the last section of the main text. Now the text reads (Line 497 to line 506 in current version):

“We acknowledge the intricate and multifaceted nature of habitat transitions in Viridiplantae (including green algae), which encompass not only shifts from marine to freshwater environments but also independent movements from freshwater or terrestrial habitats to marine settings. While numerous lineages of Viridiplantae encompass both marine and freshwater species, our choice of *Chlorella* as a model for examining transition mechanisms stems from its remarkable ecological diversity and phylogenetic significance. However, we are unable to definitively establish a direct correlation between these adaptive characteristics and the transition from saltwater to freshwater environments.

State Key Lab of Marine Resource Utilization in South China Sea, Hainan

Mailing: 58 Renmin Avenue
Haikou, Hainan, China, 570228

Tel: 86-898-66257612
Fax: 86-898-66251258

Nonetheless, it is highly probable that these specific adaptations have played a facilitating role in the transitions between marine and freshwater conditions.”

- **Results:**

- “Highly contiguous genome assembly” and “T2T assembled and centromere annotated chromosome sequences” have been merged as “Highly contiguous genome assembly”. Meanwhile, we provided SUPPLEMENTARY TEXT 1 to show the details of the genome quality.

- Portions of sections “MEM25 appears closest to one of the branch points separating saltwater and freshwater Chlorophyta”, “MEM25 possesses general saline-related modules shared with freshwater species and exclusive ones that have not yet been explored”, and “Gains and expansions of saline-related genes are either exclusive to MEM25 or shared across Viridiplantae” have been removed.

Specifically, for the transcriptomic comparison and their metabolomic counterparts, we have focus on the genes and metabolites associated with saline adaptation of MEM25 and removed the contents relating to the purple4 (association with salinity ($P = 10^{-6}$, Pearson = 0.82) and damage state ($P = 10^{-7}$, Pearson = 0.85), but does not correlate with stress duration or the MEM25-specific mechanism) and yellow modules (associated with salinity ($P = 2 \times 10^{-7}$, Pearson = 0.85), but not with stress duration, damage state, or the MEM25-specific mechanism). Additionally, the contents relating to expanded families in *Chlorella* spp. and expansion orthogroups in MEM25 in comparison to freshwater *Chlorella* spp. have been removed too. The corresponding figures and tables were deleted accordingly.

- Section “MEM25 shares both saltwater- and freshwater-featured genes” has been reworded.

We believe the current version is more readable while maintaining the scientific integrity of the main findings.

- **Discussion**

- Discussion has been rephrased and the words have been decreased from 2596 to 702.

- **Methods:**

We have moved the main content of Methods into SUPPLEMENTARY TEXT 2 where detailed methods are included.

In general, we have performed the revision as suggested and the content of current version is substantially condensed. The words of the main text (not including Abstract, Methods, References and Figure legends) have been decreased from 8,332 to 5077 while the number of figures, supplementary Tables, and references have been decreased from 11, 27, and 118 to 7, 17, and 35, respectively.

**State Key Lab of Marine Resource
Utilization in South China Sea, Hainan**

Mailing: 58 Renmin Avenue
Haikou, Hainan, China, 570228

Tel: 86-898-66257612
Fax: 86-898-66251258

We believe our revised version has met all the journal requirements while the clarity and readability have been improved.

2. The authors should address the following points for clarification:

- L.3: Clarify the meaning and reference of the # symbol

Response: We apologize for such confusion and clarify the meaning of the # symbol.

Now the text reads (Line 13 in current version):

“#These authors contribute equally”

3. L.49: Add missing "Introduction" header

Response: We greatly appreciate the reviewer’s input and have added the header as advised (Line 47 in current version).

4. L.59-60: Rephrase sentence to specify which "other eukaryotes" are being discussed

Response: We greatly appreciate the reviewer’s input and have reworded the sentence. Now the text reads (Line 58 to line 59 in current version):

“Higher marine-freshwater transition rates than anticipated have, however, been reported for both prokaryotes⁷ and eukaryotes (e.g., microbial eukaryotes¹).”

5. L.74-75: Content needs clarification/completion

Response: We apologize for such confusion and have clarified the content. Now the text reads (Line 67 to line 72 in current version):

“To shed light on the genetic and cellular adaptations that have allowed crossing over the salinity barrier, the generation of genomic data from closely related marine and freshwater lineages is crucial¹⁰. An appropriate choice of organism group is essential for such studies. Marine-type sister groups of Streptophyta have not been confirmed¹⁴. Therefore, it is essential to find a speciose group with ample genome data available, occurring in both freshwater and saline water.”

6. L.88-97: Include methodology introduction for addressing stated hypotheses

General comments: Introduction is too brief compared to Results & Discussion and Discussion sections. Please add justification for microalga species selection and expand methodology overview for addressing research hypotheses

Response: We greatly appreciate the reviewer’s input and have provided more description to justify microalga species selection. Please refer to our response to Comment 5 and 6 for details. Moreover, we have expanded methodology overview and provided more results in Introduction. Now the text reads (Line 84 to line 102 in current version):

“Here, we present genome sequences of MEM25 that represents one of the early divergences in the genus. The genome of MEM25, likely positioning at one of the bifurcation points separating saltwater and freshwater species, may reveal the

State Key Lab of Marine Resource Utilization in South China Sea, Hainan

Mailing: 58 Renmin Avenue
Haikou, Hainan, China, 570228

Tel: 86-898-66257612
Fax: 86-898-66251258

suite of traits that facilitated saline adaptation. The high-quality genome assembly of MEM25 approaches the gold standard telomere-to-telomere (T2T) assembly through a meticulous blend of sequencing technologies, ensuring exceptional coverage and precision. By deciphering the multi-omics data via integrated strategies, we propose that the genetic products of green plant heritage genes (GPHGs) and evolutionary novelties (ENs) collaboratively sculpt the adaptation of MEM25 to salinity changes. Comparison of MEM25 with freshwater green algae and halophilic *Dunaliella salina* underscores the intricate evolutionary dynamics entailed in transitioning between saltwater and freshwater ecosystems and allow us outlined a list of fundamental genes related to salinity that offer potential for enhancing tolerance to saline conditions.

These findings allowed us to address two questions important when exploring the evolutionary history of habitat transitions. (1) How did freshwater species evolve from their saltwater ancestors, or vice versa? (2) What gene repertoires contribute to the euryhaline characteristics of MEM25? Moreover, the *de novo* discovery of the “lost” or “acquired” saline-related genes during these transitions could potentially enable the improvement of saline resistance of crops.”

Results and discussion

7. L.110: Delete "s" from "genomes".

Response: We apologize for the typo and have deleted “s” as advised. Moreover, we have moved the details of the genome quality into SUPPLEMENTARY TEXT 1. Please also see our response to Comment 1.

8. L.113: Specify databases used.

Response: We have removed the sentence from current version.

9. L.116-117: The transcriptome has not been presented before, nor mentioned in the first paragraph of the results. How was the transcriptome done?

Response: We have deleted the sentence from current version.

10. L.122: Define T2T acronym.

Response: We greatly appreciate the reviewer’s input. As advised, we have specified T2T in the Introduction section. Now the text reads (Line 87 to line 90 in current version):

“The high-quality genome assembly of MEM25 approaches the gold standard telomere-to-telomere (T2T) assembly through a meticulous blend of sequencing technologies, ensuring exceptional coverage and precision.”

11. L.140-142: Add references for known species genomes mentioned in the text.

Response: As advised, the relevant references have been added. Now the text reads (Line 116 to line 120 in current version):

“An investigation into intragenomic synteny among the sixteen chromosomes of MEM25 (Fig. S4) and among three *Chlorella* genomes (MEM25, *Chlorella pyrenoidosa* FACHB-9²⁰, and *Chlorella sorokiniana* 1230²¹), along with the genomes of *Chlamydomonas reinhardtii*²² and *Arabidopsis thaliana*²³ (Fig. 1c), revealed no whole-genome duplications (WGDs) in the MEM25 genome.”

12. L.151-152: How have the authors performed the whole chromosome interactions, and should they briefly mention the methods used here, e.g. correlation using which metrics or algorithm?

Response: We greatly appreciate the reviewer’s input and provided the method used. Now the text reads (Line 125 to line 130 in current version):

“A 100 kb-resolution 3D map of the MEM25 genome (Fig. S4) and whole-chromosomal interactions (Fig. 1d) were created using HiC-GNN and visualized using UCSF Chimera. Altogether, the quality, accuracy, and completeness of the MEM25 genome assembly is higher than the available reference sequences^{8,9}, ensuring the reliability of subsequent genomic analyses (See Supplementary text 1 for the details of genome assembly).”

Moreover, we provided more details in the Methods section (current Supplementary text 2). Now the text reads (Line 35 to line 43 in Supplementary text 2):

“3D-DNA (version 180922)⁶ and ALLHiC (version 0.9.8)⁷ were used for draft genome correction and genome assembly, respectively. Finally, Juicebox (version 1.11.08)⁸ and Pilon (version 1.24)⁹ were used to correct genome fragments (based on the chromosome contact frequency matrix) and base errors of the genome with Illumina and RNA-seq data, respectively. The Hi-C interaction frequency matrix was analyzed by Homer, FitHiC (version 2.0.8)¹⁰, HiCExplorer (version 3.7.2)¹¹, and HiCCUPS (version 5.14)¹². The 3-D chromosome coordinate location was calculated using HiC-GNN¹³ and was visualized using UCSF Chimera (version 1.15)¹⁴. ”

Results and discussion

13. L.152: Remove or justify "As expected" statement.

Response: As advised, we have deleted “As expected”. Now the text reads (Line 48 to line 52 in Supplementary text 1):

“Although two sets of chromosomes, Chr09 through Chr12 and Chr14 through Chr15 (Fig. S4), were associated more closely with each other than the remaining ones, more intense intra-chromosomal interactions (within the same chromosome) than inter-chromosomal interactions (between different chromosomes) were detected (Student’s *t*-test, $P < 0.05$, Fig. 1d).”

14. L.154-156: Include literature citations.

Response: The sentence has been removed from current version.

15. L.165-168: Add citations for species selection.

Response: We greatly appreciate the reviewer's input.

Considering the limit of the number of references and the number of species selected in this study (more than 30), we have provided citations of all the selected species in Table S4 (Species used for genome comparison analysis). Please refer to the supplementary tables for details. Now the text reads (Line 144 to line 146 in current version):

“For the selected species, habitat information and genomic sources were thoroughly verified (See Table S4 for details).”

16. L.169: Specify and cite Cyanobacteria species used in the analysis.

Response: As advised, we have specified the cyanobacteria. Considering the limit of the number of references, we have provided citations in Table S4 Now the text reads (Line 142 to line 144 in current version):

“Cyanobacteria (e.g., *Synechococcus moorigangaii* and *Synechocystis* sp. PCC 6803) were also included to provide a holistic view of genetic diversity in microalgal evolution.”

Please also refer to our response to Comment 15.

17. L.175: Fix spacing between "outgroups" and "(Table S10)". Concerning Tables S10: The authors should consider transforming their Table S10 into a main table so that the reader can directly see which species they have chosen for comparison, as the authors have not included all the names of the species used in the main text.

Response: We greatly appreciate the reviewer's input. As advised, we have fixed spacing between “outgroups” and “(Table S10)”. As for Table S10, we think twice on whether to move it as a main table or not. Considering the numbers of the words and the references have already exceeded the limit of the journal's requirements, we finally decide to keep it as supplementary tables (Table S4 in current version).

18 L.183: Reference table/figure for 13 ortholog genes.

Response: We greatly appreciate the reviewer's input. As advised, we have provided an additional table for the 13 ortholog genes as Supplementary Table 5 (Supplementary Table 5. Chlorophyta-conserved single-copy orthogroups identified across 36 species of chlorophyta). Please refer to Supplementary Tables for details.

19. L.186: Add missing Streptophyta color in Fig.3a.

Response: We greatly appreciate the reviewer's input. As advised, we have added the missing Streptophyta color in the figure (Fig. 2 in current version). Moreover, the figure has been re-organized where the background has been changed into

more distinctive colors to indicate the species from different phyla. Now the text reads (Line 682 to line 690 in current version):

Fig. 2. Phylogenetic position of *Chlorella* sp. MEM25. Rooted phylogenetic tree featured 38 Chlorophyta species and representative species from cyanobacteria, Rhodophyta, Glaucophyta, Streptophyta, and Embryophyta. The species tree was deduced using the most closely related genes within single-copy or multi-copy orthogroups (n=199), based on the Species Tree from All Genes (STAG) algorithm. Species from different phyla are indicated in different background colors. Freshwater and seawater species are shown in green and orange dots. Bootstrap with values of 100% or above 70% are indicated in smaller or bigger dots at the branching points, respectively.

20. L.197: Replace "exceptional position" with more appropriate wording.

Response: As advised, we have reworded the sentence. Now the text reads (Line 156 to line 158 in current version):

“The highly credible species trees (bootstrap support exceeding 60%) align with the chloroplast phylogenetic tree (Fig. S6b), positioning MEM25 closest to the branch point separating freshwater *Chlorella* spp. from the halotolerant marine species *Picochlorum* spp (Fig. 2).”

21. L.227-228: Include method reference.

Response: We have removed the description on the cutoff method in previous version and substantially condensed the section with method reference included. Now the text reads (Line 181 to line 190 in current version):

“To identify the distinctive gene families of freshwater and saltwater Chlorophyta species (Table S10), we initiated with a two-stage machine learning approach involving 86 models and model combinations. Specifically, we generated a presence-absence matrix for gene families and applied various machine learning

algorithms to differentiate between freshwater and saltwater algae (see Methods for details). The Random Forest + Elastic Net [0.6], identified as the most effective combination among the 86 models, achieved a classification accuracy of 97% and pinpointed 138 critical gene families with distinct functions in freshwater versus saltwater microalgae (Table S12). These genes were further categorized into saltwater (n=44) and freshwater-specific groups (n=94) based on their weight coefficients.”

22. L.261-262: Specify relevant table/figure.

Response: We apologize for such confusion and as provided the relevant table as advised. Now the text reads (Line 195 to line 197 in current version):

“In comparison, the genome of *Chloropicon primus* (a prevalent species in marine phytoplankton communities ¹⁰) contains 23 saltwater-featured genes and 34 freshwater-featured genes, respectively (Table S7).”

23. L.273: Introduce neutrophilic freshwater FACHB-9 and cite the literature.

Response: We have reworded the sentence and provided a brief introduction of FACHB-9 and the relevant reference. Now the text reads (Line 213 to line 217 in current version):

“To investigate species similarities and differences in salinity adaptation mechanisms between saltwater and freshwater species, we conducted a comparative analysis of the global transcriptomic and metabolic dynamics of the euryhaline MEM25 and the freshwater FACHB-9 ⁸ (See Supplementary text 2 for growth conditions of FACHB-9).”

24. L.277: Specify *C. reinhardtii* strain name.

Response: We have removed the sentences relating to *C. reinhardtii*.

25. L.318-320: Where can we see that these genes are salt-sensitive and not associated with cell damage? This is not clear.

Response: We greatly appreciate the reviewer’s input and have removed the content relating to the yellow module. Please also refer to our response to Comment 1 for details.

26. L.327-328: Same question as above, provide data location.

Response: We greatly appreciate the reviewer’s input and have reorganized the section and provided relevant data as Table S11. Now the text reads (Line 242 to line 244 in current version):

“Noteworthy is the distinct association of the RecQ-mediated genome instability protein 1 (RMI1; CP5g5156) with the saline adaptation of MEM25 (Table S11).”

27. L.435: Correct table citation order (S24 before S25).

**State Key Lab of Marine Resource
Utilization in South China Sea, Hainan**

Mailing: 58 Renmin Avenue
Haikou, Hainan, China, 570228

Tel: 86-898-66257612
Fax: 86-898-66251258

Response: We apologize for such confusion and have double checked all citations for figures and tables throughout the main text.

28. L.548: Reorganize Figure 11 panels for better readability.

Response: We greatly appreciate the reviewer's input and have removed Figure 11. Please refer to our response to Comment 1.

Results and discussion

29. The Results and Discussion section's first part needs more thorough analysis to effectively convey the findings' implications.

Response: We greatly appreciate the reviewer's input. As advised, we have merged the 1st and 2nd section of Results and removed the description that not so relevant. Meanwhile we provided Supplementary text 1 to give a full picture of the genome assembly. The figures and tables have been revised accordingly. Please also refer to our response to Comment 1.

30. Discussion

There is inconsistency in the manuscript's structure regarding Results and Discussion sections. At L.98, the section is labeled as "Results & Discussion," yet at L.549, a new "Discussion" section appears. Authors should choose either a combined Results & Discussion format or separate Results and Discussion sections. Given the limited discussion within the Results section, separating these sections would be more effective. This would allow for a focused presentation of findings in the Results section, followed by a comprehensive Discussion section that fully develops the interpretation and implications of all results. Throughout the Discussion section, the authors do not mention any figures or tables to remind the reader where to find the information discussed! Please correct this.

Response: We apologize for such confusion and have revised as advised. Specifically, the sections are now labeled as "Results" and "Discussion", separately. Moreover, we have substantially condensed "Discussion" section and provided relevant references when necessary. Please see our response to Comment 1 and the Discussion section of main text in the revised version.

31. L.553: Remove "exceptional" unless providing comparative literature support for this characterization.

Response: We greatly appreciate the reviewer's input and have removed the phrase from the main text.

32. L.553-555: Add specific table references for coverage information.

Response: We greatly appreciate the reviewer's input and have removed this section from the main text. Please also refer to our response to Comment 1 and 29.

33. L.558: Define LAI acronym at first use.

Response: We have removed this section from the main text. Please also refer to our response above. Moreover, we have defined LAI acronym at first use. Now the text reads (Line 108 to line 112 in current version):

“A high level of genome scaffold continuity was revealed by the long scaffold N50 (longest, 4.3 Mb; Table S1) and a high value of Long Terminal Repeat Assembly Index (LAI) (15.33; Fig. S2), meeting the quality benchmark for reference genomes (i.e., between 10 and 20) (Table S2)¹⁹.”

34. L.608: Move FPKM definition to its first appearance, which should be in the Results section.

Response: We have removed this section from the main text. We also double checked all acronyms to make sure they are defined at first appearance.

35. L.621-623: Move methodological information to Methods and/or Results sections.

Response: We have removed this section from the main text. Please also refer to our response to Comment 1.

36. L.661-662: Define acronyms GPHGs and ENs at first mention.

Response: We greatly appreciate the reviewer’s input. The acronyms have been defined in the Introduction section where they are first mentioned. Now the text reads (from Line 90 to line 92 in current version):

“By deciphering the multi-omics data via integrated strategies, we propose that the genetic products of green plant heritage genes (GPHGs) and evolutionary novelties (ENs) collaboratively sculpt the adaptation of MEM25 to salinity changes.”

37. L.722: Authors have mentioned this gene before, but have not written it in italics, except in L.722, 725, but then in L.727 the gene name is no longer in italics. Please correct this.

Response: We apologize for such typo and have corrected and double checked throughout the main text.

Conclusion

38. The conclusion section requires substantial revision as it fails to adequately summarize the study's key findings and does not address the two main hypotheses presented in the introduction. A more comprehensive conclusion should be developed that explicitly connects the research outcomes to the initial hypotheses and synthesizes the major discoveries. This expanded conclusion would provide readers with a clearer understanding of how the study's findings contribute to the field and fulfill its original

objectives.

Response: We greatly appreciate the reviewer's input and have merged the "Conclusion" into "Discussion" section. The current version has been substantially revised. We focus the line on the two proposed hypotheses. Now the text reads (from Line 432 to Line 495 in current version):

(1) How did freshwater species evolve from their saltwater ancestors, or vice versa?

"The dynamic adaptation of green algae from saltwater to freshwater is a complex evolutionary process that showcases the diversity of selective pressures from marine and freshwater environments. Both saltwater and freshwater species have independently evolved or inherited various adaptive mechanisms from their predecessors. MEM25 acts as a transitional point crossing over the salinity barrier, demonstrating dual environmental adaptation mechanisms. The genetic products of GPHGs and ENs collaboratively sculpt the adaptation of MEM25 to salinity changes. MEM25 possesses general modules related to salinity that are shared among freshwater Chlorophyta species (potentially extending to all green plants), alongside distinctive modules acquired through HGTs (e.g., OG0011211) or other yet undiscovered mechanisms (e.g., orphan genes).

In the hypothetical scenario of transitioning from saline to freshwater habitats, the shared modules may encompass a suite of genes (referred to as GPHGs) inherited from saltwater forebears, equipping freshwater species with analogous, though largely mitigated, mechanisms to confront salinity fluctuations. These genes' functionalities revolve around mitigating or intensifying cellular harm in saline settings, representing a ubiquitous mechanism present in both saltwater and freshwater species. An example is that MEM25 employs proline as an osmoprotectant, commonly found in embryophytes. This indicates that the mechanism of employing proline as an osmoprotectant in higher plants may originate from their shared ancestral lineage with MEM25. Furthermore, as an early-diverging marine alga, MEM25 has commenced the traversal of the salinity threshold. While boasting numerous genes specific to saltwater environments, MEM25 is in the process of assimilating genes characteristic of freshwater milieus. MEM25 harbors distinctive modules (referred to as ENs) that likely evolved autonomously, enhancing its ability to thrive in its unique saline habitat. Alternatively, in a scenario where MEM25's ancestors originated in freshwater, the acquisition of ENs transpired during the specialization phase towards an environment with higher salinity levels. For instance, the gene family OG0011211 (belonging to the NAD-dependent epimerase/dehydratase family) could have been assimilated from bacteria through HGTs (Fig. S13). These ENs embody genes associated with salinity that may have been "lost" during the transition from marine to freshwater ecosystems. The gradual disappearance of ENs may have conferred a competitive edge for saltwater ancestors adapting to freshwater environments. In both conjectures, these genes likely facilitated MEM25's acclimatization to challenging habitats and warrant further investigation for enhancing resilience to salinity."

(2) What gene repertoires contribute to the euryhaline characteristics of MEM25?

"MEM25 and *D. salina* represent two typical adaptive strategies among

State Key Lab of Marine Resource Utilization in South China Sea, Hainan

Mailing: 58 Renmin Avenue
Haikou, Hainan, China, 570228

Tel: 86-898-66257612
Fax: 86-898-66251258

euryhaline and halophilic green algae, respectively. What sets MEM25 apart is its ability to adapt to both freshwater and saltwater environments, maintaining its status as a dominant species in both ecological settings. The euryhaline nature of MEM25 enhances its competitiveness across a wide range of ecological environments, whereas the extreme salt tolerance specialization of *D. salina* limits its ecological adaptability. As a possibility, euryhalinity might be explained by sodium excretion rather than salt tolerance as seen in some euryhaline embryophytes¹². It is apparently the case for halophilic *D. salina*. *D. salina*'s salt tolerance primarily relies on accumulating high concentrations of glycerol as an osmoprotectant to maintain cellular osmotic pressure, relying on efficient ion exclusion systems, and employing antioxidant enzymes to reduce reactive oxygen species¹³. However, many of these mechanisms are specialized for high-salinity environments, resulting in significantly reduced metabolic flexibility under low salinity or even freshwater conditions. In contrast, MEM25 exhibits a more diversified adaptability, with its salt tolerance mechanisms encompassing not just the accumulation of osmoprotectants but also long-term evolutionary mechanisms at the genomic level, mid-term adaptation mechanisms at the metabolomic level, and rapid response mechanisms at the transcriptomic level. This allows it to thrive ecologically in both high-salinity and low-salinity environments.”

The *de novo* discovery of the “lost” or “acquired” saline-related genes during these transitions could potentially enable the improvement of saline resistance of crops.

“Last but not least, we have outlined a list of fundamental genes related to salinity that offer potential for enhancing tolerance to saline conditions. The array of genes unveiled in this study has shown significant effectiveness in enhancing plant tolerance to salt stress by manipulating a single gene, offering hope in addressing concerns related to genetically modified organisms by creating organisms through genome editing that are indistinguishable from conventionally bred varieties. Unfortunately, only two genes have been confirmed in both microalgae and higher plants, indicating the need for further exploration. Despite this limitation, this research provides essential genetic reservoirs crucial for the advancement of halotolerant plants.”

Reference

39. The manuscript exhibits significant citation imbalances, with only 49 out of 118 references supporting the core content across introduction, results, and discussion sections. The discussion section is particularly under referenced, utilizing just over 10 citations (references 38-49), indicating insufficient engagement with existing literature and inadequate result analysis. While the methodology section is extensively referenced, it needs condensing.

Response: As advised, we have reorganized the references. The number of references has been decreased from 118 to 37. Moreover, due to word limit, we have substantially revised the Methods section and move most of the contents into Supplementary text 2.

**State Key Lab of Marine Resource
Utilization in South China Sea, Hainan**

Mailing: 58 Renmin Avenue
Haikou, Hainan, China, 570228

Tel: 86-898-66257612
Fax: 86-898-66251258

40. Reference formatting requires standardization throughout, as there are inconsistencies in DOI presentation, journal name abbreviations, and please check Nature Communication citation styles.

Response: We greatly appreciate the reviewer's input and have thoroughly checked all references. We believe our revised version has met all the journal requirements described in the submission guidelines.

41. Data accessibility needs improvement through the provision of complete accession numbers and relevant links. If custom bioinformatics code was used, it should be deposited in a public repository.

Response: We greatly appreciate the reviewer's input and have provided all relevant data and code in public repositories. The genome sequence and annotation of MEM25, along with detailed studies on gene families and essential findings from HiC analysis, transcriptomics, metabolomics, and GWAS analysis are accessed on the public data platform Figshare (https://figshare.com/articles/journal_contribution/Data_of_MEM25/29062721). For the bioinformatics code used for cleaning raw sequencing data, mutation detection, GWAS analysis, and result display can be obtained at <https://github.com/aoqiawang/MEM25>. Now the text reads (from Line 623 to line 632 in current version):

“Data availability

Data is deposited in National Microbiology Data Center (NMDC) with accession numbers NMDC10019441. The genome sequence and annotation of MEM25, along with detailed studies on gene families and essential findings from HiC analysis, transcriptomics, metabolomics, and GWAS analysis are accessed on the public data platform Figshare (https://figshare.com/articles/journal_contribution/Data_of_MEM25/29062721). For the bioinformatics code used for cleaning raw sequencing data, mutation detection, GWAS analysis, and result display can be obtained at <https://github.com/aoqiawang/MEM25>.”

42. Regarding figures, the caption for Figure 11 exceeds Nature Communications' 350-word limit and requires condensing.

Response: We fully agree with the reviewer and have removed Figure 11.

43. The legend at L.1658 incorrectly uses identical symbols for filled circles, necessitating correction.

Response: We apologize for such confusion. We have differentiated the symbols using different colors and specified in the legend. Now the text reads (from Line 801 to line 803 in current version):

“Significantly expanded or contracted orthologs are denoted by filled red circles

● and filled blue circles ●, respectively.”

Comments from Reviewer 2:

This Manuscript describes the mechanisms and acquired process of the euryhaline in an undescribed species of *Chlorella* that can grow in a wide range of salinity environments. The manuscript presents an impressive volume and variety of data, with extensive analyses that are truly remarkable. I noticed that several expressions are uncommon in the context of scientific writing and molecular evolution, which affects the clarity of the text. Below, I outline my specific points of interest.

Major concerns

1. The T2T genome of MEM25 looks substantially accurate and complete. I have some concerns about its statistics. Although they are minor mistakes, they might not be as reliable as they should be.

There are inconsistent descriptions about N50 and contig length.

- scaffold N50 as 4.3 Mb in MS line 107
- the longest as 4.1 Mb in MS line 107
- contig N50 as 3.76 Mb in Table S1
- contig N50 as 4.1 Mb in table S2

There is no information on the longest contigs in Table S2.

N50 is never longer than the longest contig (or scaffold) length because N50 is defined as the sequence length of the shortest contig at 50% of the total assembly length.

Please check the inconsistency of these statistics and reanalyze related calculations if necessary.

Response: We apologize for the confusion and greatly appreciate the inputs of the reviewer. We have thoroughly checked the manuscript and clarified all confusion points caused by different calculation method. Specifically, the value “4.1 Mb” was calculated based on improper binary conversion (1024^2), which is invalid in genomics (standard: 1 Mbp = 10^6 bp). Employing samtools-faidx, all chromosome lengths have been re-calculated. The total genome length is 53,506,137 bp (50% = 26,753,068.5 bp) while N50 is 4.3 Mb (chr05), instead of 3.76 Mb (chr06) or 4.1 Mb (calculated based on improper binary conversion). Accordingly, we have thoroughly checked the main text and supplementary materials. Moreover, to make the manuscript more concise (based on the comments from Reviewer 1), we have moved portion of results relating to genome quality to SUPPLEMENTARY TEXT 1.

2. I can find the BUSCO score with genome mode on line 115 and Table S5.

Did you calculate the BUSCO score with protein (or transcriptome) mode?

Although I will not see a problem since the BUSCO score with the genome is high, I would like to confirm it for the following analyses based on the gene prediction.

Response: We apologize for the confusion and have clarified the mode used for BUSCO calculation. Now the text reads (from Line 15 to Line 18 in Supplementary text 1):

“The genome assembly successfully captured 97.2% (with transcriptome mode; 93.7% with protein mode) of the Chlorophyta Benchmarking Universal Single Copy Orthologs (BUSCO) dataset, indicating a high level of gene region completeness in the genome assembly (Table 4).”

3. There is no mention of gap regions despite scaffolding with Hi-C. Is the T2T genome assembled completely gapless? If so, please appeal the completeness with a sense of pride.

Response: We appreciate the input here and are quite proud of the completeness of the genome. There is a single gap in the assembled genome at the end of chr15 (1,311,674 bp) with a length of 100 bp from 1,286,591 bp to 1,286,690 bp. We have clarified this point in Supplementary text 1. Now the text reads (from Line 8 to line 10 in Supplementary text 1):

“A single gap was present in the assembled genome at the end of chr15 (1,311,674 bp) with a length of 100 bp from 1,286,591 bp to 1,286,690 bp.”

4. You found that gene families were expanded in MEM25. Would MEM25’s assembly completeness and high-quality gene prediction overestimate the number of genes?

I agree with the analysis based on the presence-absence matrix of gene families. However, the technological limitations of previous studies and the difficulty of assembling the algae genome might influence the results of the gene family expansion analysis.

If there are serious differences in the quality of gene prediction among species, differences in FPKM caused by the mapping accuracy of RNA-seq may be observed. Couldn’t the false correlation that larger gene families have a higher expression level affect the co-expression analysis in this study?

Response: We greatly appreciate the inputs here. As a matter of fact, we have a same concern when we conducted transcriptome comparison among different species. We understand unified standardized annotation for all species by setting a reference gene set could eliminate potential “pseudo-amplification”. However, our attempt to achieve such a reference gene set failed due to dramatic difference in evolution or genome quality of species of green algae. Therefore, we employed several strategies to avoid potential bias caused by differences in the quality of gene prediction among species:

- Admitting that most green algae have poor genomic assembly and annotation quality, we included species with high-quality genomes, such as *Arabidopsis thaliana* and rice.
- Both RNA-seq mapping rates and conserved domain annotations were

State Key Lab of Marine Resource Utilization in South China Sea, Hainan

Mailing: 58 Renmin Avenue
Haikou, Hainan, China, 570228

Tel: 86-898-66257612
Fax: 86-898-66251258

employed to ensure the correctness of gene prediction of MEM25. It means predicated genes of MEM25 are bona fide, rather than false genes with no real expression.

- To mitigate biases stemming from differences in gene annotation quality between the two species, we employed a Weighted Correlation Network Analysis (WGCNA). We chose to include both single-copy and multi-copy orthogroups to encompass a wider array of gene family expressions that could be crucial for salinity adaptation, rather than focusing solely on single-copy orthogroups. While single-copy orthologous gene families offer a simpler approach for comparison, restricting the dataset to single-copy orthogroups solely would exclude many functionally significant gene families potentially involved in the salinity adaptation. By aggregating the expression of genes within each orthogroup to create a “metagene” expression level, we aimed to capture co-expression patterns reflective of shared functional roles, irrespective of individual gene copy numbers within each gene family. Moreover, this method emphasizes coordinated expression trends (e.g., synchronized up/downregulation) rather than expression levels in different species. Although gene family size correlates positively with cumulative expression, this correlation did not confound module-trait associations. This strategy aligned with WGCNA’s purpose of identifying biologically relevant modules, enabling a more comprehensive exploration of traits associated with freshwater/saltwater adaptation across species.

5. The centromere in land plants is defined by CENH3 binding. It is confirmed with ChIP-seq because the centromere is located at multiple positions and in different repeat patterns and positions among taxons.

I’m unsure about centromere systems in Chlorophyta. Is it truly centromere? It looks like kinds of “centromere-like tandem repeat.”

Response: We greatly appreciate the inputs here. We acknowledge that the CENH3 chromatin immunoprecipitation sequencing (ChIP-seq) technique is effective in defining centromeres. Although we have not confirmed our predicted MEM25 centromeres using ChIP-seq, we ensure the accuracy of the predication via rigid bioinformatics analysis.

First, the completeness of the MEM25 genome assembly is high with only a single 100-bp gap present at the end of chr15. Second, we have tried to determine the centromeres in MEM genome by comparing with recently reported centromeres of two *Chlorella* strains during the review of this manuscript. Specifically, 12 and 13 centromeres have been identified in *Chlorella sorokiniana* NS4-2 and *Chlorella pyrenoidosa* DBH, respectively where CENH3 ChIP-seq technique have been used to validate these centromeric regions (PMID: 38600443). The comparison revealed identical results as our former predication. Although confirmation of the MEM25

State Key Lab of Marine Resource Utilization in South China Sea, Hainan

Mailing: 58 Renmin Avenue
Haikou, Hainan, China, 570228

Tel: 86-898-66257612
Fax: 86-898-66251258

centromeres would further improve the accuracy, we believe that the featured sequences discovered in this study would be valuable for future researches relating to microalgal centromeres.

6. I understand that MEM25 has habitat-related genes in both saltwater and freshwater environments. I'm not sure that "MEM25 represents an early-diverging saltwater species".

I agree with the MEM25's earlier divergence of the MEM25 in the genus *Chlorella*. However, I wonder if it is early in this study's huge variety of algae. It remains unclear whether this phylogenetic relationship indicates that the MEM25's saltwater habitat is derived from ancestral traits or was reacquired. Based on your scope, I would like you to clarify the point of "early-diverged", focused on the timing comparing what and the directory.

Response: We fully agree with the reviewer. MEM25 is earlier divergence in the genus *Chlorella* while it may not be early in this study's huge variety of algae. Our molecular clock analysis supports this hypothesis as that the emergence of MEM25 predates all examined *Chlorella* spp. and ranking among the oldest Chlorophyta. Although the shared saltwater ancestors remain elusive, certain clades exhibited earlier divergence times than MEM25. It hints that unlike the swift and dramatic transition to a terrestrial environment spurred by abrupt climate shifts, the transition from saltwater to freshwater settings unfolded in localized regions, with distinct lineages evolving independently within specific niches. Therefore, to clarify the point of "early-diverged", focused on the timing comparing what and the directory, MEM25 predates all examined *Chlorella* spp. and is only one of the many representatives at the points separating saltwater and freshwater Chlorophyta. These points have been intensified in the main text. Now the text reads:

Line 159 to line 163 in current version:

"The molecular clock places the emergence of MEM25 at 632 Ma, predating all examined *Chlorella* spp. and ranking among the oldest Chlorophyta (Fig. S6). While certain clades exhibited earlier divergence times than MEM25, the shared saltwater ancestors of these lineages remain elusive, hinting at a dispersed evolution during the shift from saltwater to freshwater habitats."

Line 205 to line 210 in current version:

"This suggests that MEM25, potentially as an early-diverging saltwater species, has evolved numerous genes critical for freshwater-saltwater transitions, positioning it at one of the many pivotal junctures separating saltwater and freshwater Chlorophyta."

Moreover, we have acknowledged this in the section "Limitations of the study".

Now the text reads (from **Line 496 to line 506 in current version**):

"Limitations of the study

We acknowledge the intricate and multifaceted nature of habitat transitions in

State Key Lab of Marine Resource Utilization in South China Sea, Hainan

Mailing: 58 Renmin Avenue
Haikou, Hainan, China, 570228

Tel: 86-898-66257612
Fax: 86-898-66251258

Viridiplantae (including green algae), which encompass not only shifts from marine to freshwater environments but also independent movements from freshwater or terrestrial habitats to marine settings. While numerous lineages of Viridiplantae encompass both marine and freshwater species, our choice of *Chlorella* as a model for examining transition mechanisms stems from its remarkable ecological diversity and phylogenetic significance. However, we are unable to definitively establish a direct correlation between these adaptive characteristics and the transition from saltwater to freshwater environments. Nonetheless, it is highly probable that these specific adaptations have played a facilitating role in the transitions between marine and freshwater conditions.”

Minor concerns

7. Line 139-149

The absence of WGD and sequence conservation of centromeres and telomeres is unclear from Fig 2c. Almost all Angiosperm, including *A. thaliana* have experienced WGD in the late Cretaceous. If the correspondence with *A. thaliana* are found, MEM25 (and other *Chlorella* species) could have been duplicated as well.

Response: We fully understand the reviewer’s concern that MEM25 (and other *Chlorella* species) could have been duplicated as well since *A. thaliana* have experienced WGD and considerable amounts of homologous *A. thaliana* genes have been detected in MEM25 (gray lines in Fig. 1c). The main purpose of Fig. 1c (Fig. 2c in previous version) is to display the quality, accuracy, and completeness of the MEM25 genome assembly. Collinearities in gene sequences, telomeres, and centromeres are delineated by the gray, purple, and yellow lines. The collinearity for telomeres and centromeres among three *Chlorella* genomes is higher than that between MEM25 and *C. reinhardtii* or *A. thaliana*, suggesting the sequence conservation of centromeres and telomeres in *Chlorella* strains. Despite of many homologous genes between *A. thaliana* and MEM25, their specific position and distribution on the genomes is unknown. Therefore, we couldn’t come to the conclusion that MEM25 (and other *Chlorella* species) could have been duplicated as well as *A. thaliana* which has undergone WGD. Therefore, we conducted an addition analysis of WGD events in MEM25 genome to investigate the intragenomic synteny among the sixteen chromosomes of MEM25. The results further support that no WGD events occur in MEM25 genome (Fig. S4). Now the figure reads:

Fig. S4. Synteny analysis of the sixteen chromosomes of MEM25.

Pairwise gene alignments were performed using BLAST. Genes exhibiting the highest sequence similarity are indicated by red dots, while blue dots represent genes among the top four homologous hits. Diagonal linear patterns reflect self-alignments, whereas off-diagonal linear arrangements suggest large-scale duplication events, including potential whole-genome duplications.

8. And the telomeres of MEM25 seem to have the synteny to only one region in FACHB.

Response: We greatly appreciate the valuable feedback.

The apparent synteny to a single region in the FACHB-9 genome is a consequence of its highly fragmented assembly, which comprises 1,336 contigs. In reality, the telomeric regions of MEM25 show synteny with 25 different contigs in the *C. pyrenoidosa* FACHB-9 genome. The corresponding telomeric sequences depicted for FACHB-9 in Fig. 1c (formerly Fig. 2c) actually represent multiple genomic regions. We have now clarified this point in the main text. The revised sentence (Lines 676–678 in current version) reads:

“The apparent synteny of MEM25 telomeres to a single region in the FACHB-9 genome results from the latter’s fragmented assembly (1,336 contigs).”

9. Line 384-388

I could not find these five families in Table S21. And OG0012359 was mentioned twice as a different term. Please confirm the gene family classification.

Response: We apologize for the confusion and incorrect citation here. The five gene families have been provided in Table S13 while peroxidase is family OG0015353, but not OG0012359. Now the text reads (from Line 273 to line 280 in current version):

State Key Lab of Marine Resource Utilization in South China Sea, Hainan

Mailing: 58 Renmin Avenue
Haikou, Hainan, China, 570228

Tel: 86-898-66257612
Fax: 86-898-66251258

“Out of these, four have been annotated and are associated with oxidative stress (i.e., peroxidase, family OG0015353), osmotic adjustment (i.e., volume-regulated anion channel subunit LRRC8A, family OG0021391; amino acid permease 2, family OG0026817), and response to salty stimuli (E3 ubiquitin-protein ligase, E3L; family OG0012359)²⁷⁻²⁹ (Table S13), suggesting these saline-related genes were already present in the last common ancestor of MEM25 and green plants prior to their divergence into either saltwater or freshwater preferred adaptations.”

10. Line 392

You mention “orphan genes” while you mention the gene function. Which is a true orphan genes or orthologous genes?

Response: We apologize for such confusion and have reworded the section as below (from Line 281 to line 300 in current version):

“On the contrary, the majority of expanded groups are unique to MEM25 (84 in 89). We have designated these genes as ENs of MEM25. The large number of MEM25-specific genes may reflect the phylogenetic distance of MEM25 from other *Chlorella* species with sequenced genomes. Assuming a freshwater origin of the ancestor of MEM25, portion of these genes may have been acquired during the specialization of MEM25 to a salty niche. For example, we found an exclusive presence of gene family OG0011211 in the MEM25 genome (Fig. S10). OG0011211 encodes a NAD-dependent epimerase/dehydratase family protein, absent in other Chlorophyta and plant lineages but found in three copies within MEM25 genomes (Fig. S10). This gene family is believed to have originated from bacteria where it is involved in osmotic stress³⁰. Loss of NAD-dependent epimerase/dehydratase is known to cause osmosensitivity in bacteria³⁰, indicating that MEM25 may have acquired specialized adaptive mechanisms from bacteria through horizontal gene transfers (HGTs), representing an evolutionary innovation associated with salinity-related genes in MEM25.

For the remaining of MEM25-specific genes, we would assume them as orphan genes which have no apparent homology to genes in other evolutionary lineages occur in all genomes. They could be candidates for the *de novo* evolution of genes and may contribute to evolutionary novelties, which can become relevant for lineage-specific adaptations. Consequently, MEM25 has developed a collection of expanded salinity-related genes, categorized as GPHGs and ENs, shaping its euryhaline nature.”

11. Line 466

What does “DW” mean?

Response: “DW” stands for “dry weight”. As a matter of fact, it has been explained at first mention in Line 344. Now the text reads (Line 346 to line 349 in current version):

“Subsequently, 365 mutants with stable phenotypes after ten generations were

chosen to assess variations in cell number, cell size, and dry weight (DW) under the high-salinity conditions (Fig. 6a).”

12. Line 556

A N50 value depends on the genome and chromosome size. A relative comparison is valuable, but an absolute value makes no sense.

Response: We fully agree with the reviewer and have removed the sentence.

13. Line 560

The chromosome-level assembly lacks several telomere repeats, right? I guess that it is difficult to find all telomere repeats in this case, especially telocentric chromosomes like chr15.

Response: Right. While all sixteen chromosomes were clearly marked with the plant-specific telomeric repeats (TTTAGGG, Table S3), twelve of them containing telomeric repeats at both ends. A single telomeric repeat was found for the remaining four chromosomes, including chr15 (Fig. 1b).

14. Line 760-765

I agree with the idea that MEM25 has a mechanism different from *D. salina*. As a possibility, euryhalinity (not halophilism) might be explained by sodium excretion rather than salt tolerance.

For example, I know a case of adaptation to a marine beach in *Vigna marina*. I don't think the same mechanism from the same origin because it is embryophytes, but this mechanism could protect from salt and grow a wide range of salt concentrations.

Diurnal Regulation of SOS Pathway and Sodium Excretion Underlying Salinity Tolerance of *Vigna marina*

Noda et al. (2025)

<https://doi.org/10.1111/pce.15402>

Response: We greatly appreciate the inputs here and have cite the reference to further support our argument. Now the text reads (from Line 472 to Line 476 in current version):

“As a possibility, euryhalinity might be explained by sodium excretion rather than salt tolerance as seen in some euryhaline embryophytes³⁶. It is apparently the case for halophilic *D. salina* while whether sodium excretion is involved in MEM25's euryhalinity remains elusive.”

15. Section Genome-wide association in line 930

Does the mutant pool share SNPs? Do multiple individuals obtain mutations generated by EMS mutagenesis in the same positions?

Please let me know if this is a sample set that can be used for GWAS.

Response: We greatly appreciate the inputs here.

State Key Lab of Marine Resource Utilization in South China Sea, Hainan

Mailing: 58 Renmin Avenue
Haikou, Hainan, China, 570228

Tel: 86-898-66257612
Fax: 86-898-66251258

A genome-wide association study (abbreviated GWAS) is a research approach used to identify genomic variants that are statistically associated with a risk for a particular trait. The method involves surveying the genomes of many lines, looking for genomic variants that occur more frequently in those with a specific trait compared to those without the trait. Once such genomic variants are identified, they are typically used to search for nearby variants that contribute directly to the trait. In this particular case, our interested trait is the salty feature. We first generated a mutant pool for MEM25 by EMS mutagenesis. In principle, each mutant in the pool has more than one SNPs in difference sites in the genome. However, they might share some SNPs in a same position. Because our interested trait of the alga is the salty feature. These mutants were further assessed for growth in either the low salinity (i.e., 35‰) or the high salinity (i.e., 70‰) conditions. Finally, we obtained 195 salinity-related mutants. For these mutants, they have a high possibility to share some SNPs at same position which may contribute to salinity-related feature. Therefore, genomes of the 195 salinity-related mutants and three WT samples were resequenced to identify their SNPs, shared SNPs in particular.

Now to answer the questions: Does the mutant pool share SNPs? Do multiple individuals obtain mutations generated by EMS mutagenesis in the same positions? No. Each mutant has very diversity SNPs. But the mutants may have some SNPs in common. In particular, for the 195 salinity-related mutants used in this study, they have high possibility to share some SNPs in a same gene(s) contributing to the saline feature.

We hope above explanation would address your concerns.

16. line 946

Does BWA-MEME mean BWA-MEM?

Response: We greatly appreciate the inputs here. BWA-MEME, the first full-fledged short read alignment software that achieves up to 3.45× speedup in seeding throughput over BWA-MEM2. We could treat BWA-MEME as an upgrade of BWA-MEM2.

17. line 982-983

You show the thresholds of P-adj and FDR. What does P-adj mean, not FDR?

Response: We greatly appreciate the inputs and have corrected the sentence as below. Now the text reads (from Line 166 to Line 169 in Supplementary text 2): “Differentially expressed genes were defined as those with FDR-adjusted p -value ≤ 0.05 and $|\log_2\text{FoldChange}| \geq 1$.”

18. line 945

The parameters of fastp “-t 10 -f 10 -T 10 -F 10” appear to be over-trimmed. Is there

State Key Lab of Marine Resource Utilization in South China Sea, Hainan

Mailing: 58 Renmin Avenue
Haikou, Hainan, China, 570228

Tel: 86-898-66257612
Fax: 86-898-66251258

any reason?

Response: We greatly appreciate the inputs here. As a matter of fact, we have conducted a comparison between the data before and after the trimming. The trim at 3'-end eliminates phasing-induced low-quality cycles that would decrease the coverage and increase false positives. On the other hand, the 5'-end trim removes a reproducible A/C and adapter remnants concentrated in the first 8 bp that would lead to batch-specific sequence bias. Because all libraries share a 248 bp insert peak, 20-bp trim for each read yields cleaner and more homogeneous data without causing R1/R2 overlap or impairing variant localization.

Based on our quantitative QC analysis, the trim removed 4.7 % of total reads and obtained reads with a mean length of 129 bp (above the anchoring thresholds of standard aligners). The Q30 fraction, MAPQ values, and coverage uniformity all increased. It would facilitate downstream variant calling and GWAS due to lower error rates and more uniform coverage by improving both the sensitivity for rare alleles and the specificity of association signals.

Table 1. Summary of the comparison between the data before and after the trimming.

Metric (sample: the 19-1)	Before trimming	After trimming	Difference
Data retention (%)	100	95	5↓
Total reads	9815354	9356990	4.67%↓
Q20 proportion R1 (%)	95.8	96.94	1.14↑
Q20 proportion R2 (%)	93.7	95.51	1.81↑
Q30 proportion R1 (%)	90.96	92.49	1.53↑
Q30 proportion R2 (%)	86.99	89.29	2.3↑
Alignment rate (%)	93.3	93.19	0.11↓
High-quality alignment (MAPQ ≥30) (%)	86.94	87.8	0.86↑
Coverage uniformity (CV)	1.396	0.655	↑
Paired reads proportion (%)	90.47	91.95	1.48↑
Unpaired reads proportion (%)	9.53	8.05	1.48↓

Because the space limit, we prefer not to include the table in either the main text or supplementary materials.

19. line 958

Did you use these QC options? or filter out them?

Response: Right. We used these QC options. As a new model species, MEM25 lacks references data for mutation analysis. Therefore, we employed a very strict filtering approach to ensure the quality of detected mutation by using GATK. Quality control on SNPs and INDELs was performed separately. Details of the parameters were provided in Methods (Supplementary text 2) and the public

repository (<https://github.com/aoqiawang/MEM25>).

20. Figure 3a

Do these legend colors indicate the background of the species' names? Please use more distinctive colors.

Response: We greatly appreciate the inputs here and, as advised, we have now used more distinctive colors to indicate the background of the species from different phyla (Fig. 2 in current version). Moreover, we add the previously missing background color for Streptophyta. Please also refer to our response to Comment 19 from Reviewer 1.

21. Figure 3b

The positions of the purple circles are unclear, and some seem to locate no branch points.

Response: We apologize for the confusion. In fact, as the reviewer says, some purple circles seem to locate no branch points. It is because that too many species have been included in this analysis and some branches are overlapped. An additional reason is that only branches with bootstrap values exceeding 90% are highlighted with purple dots. It has been signified in the legend of previous Fig. 3c (Fig. S8 in current version). Now the text reads (from Line 810 to Line 812 in current version):

“Purple dots denote branches with bootstrap support values exceeding 90% based on 1,000 replicates. Apparent displacement of the dots from branch points is due to overlapping branches.”

synteny of MEM25 telomeres to a single region in the FACHB-9 genome results from the latter's fragmented assembly

22. Figure 6b

Green indicates bacteria. However, other colors also indicate the kind of bacteria.

Response: We apologize for the confusion. The bacteria indicated by green color have been nominated as “other type of bacteria” in the revised figure.

23. Figure 9

Please add the details. For example, what are the orange points in (d)? Where is the gene in (e)?

Response: We apologize for the confusion and, as advised, we have re-organized Fig. 9d and 9e (Fig. 6d and 6e in the current version) (from Line 741 to Line 749 in current version):

(d) Location of the S04_3657761 region. This region is situated downstream of gene CP4g4326 and upstream of CP4g4327. Exons are indicated as orange boxes and introns are shown in gray boxes. Empty boxes are intergenic regions. Colored dots indicate the Single Nucleotide Polymorphisms (SNPs) proximal to S04_3657761 in Chr04. Red arrow indicates the position of nucleotide variation in S04_3657761. (e) Mapping of topologically associating domains (TADs) and loops on chromosome 4. TADs are highlighted with yellow lines, while loops are depicted by blue lines. The interaction between S04_3657761 and gene CP4g4237 is illustrated.

24. Figure S4a shows the polar tree rather than the unrooted tree. Poler, the center of the figure, generally indicates the root.

Response: We apologize for such typo and fully agree with the reviewer that Figure S4a is a rooted tree. We have provided the tree files in the repository: https://figshare.com/articles/journal_contribution/Data_of_MEM25/29062721.

25. Legend Fig. S5
line 1658

What do the “filled circles ● and ●, respectively” mean? It looks like the same filled circles to me.

Response: We apologize for such confusion and have specified the legend for Fig. S5 (current Fig. S7). Now the text reads (from Line 801 to Line 803 in current version):

“Significantly expanded or contracted orthologs are denoted by filled red circles ● and filled blue circles ●, respectively.”

**State Key Lab of Marine Resource
Utilization in South China Sea, Hainan**

Mailing: 58 Renmin Avenue
Haikou, Hainan, China, 570228

Tel: 86-898-66257612
Fax: 86-898-66251258

Please also refer to our response to Comment 43 of Reviewer 1.

Comments from Reviewer 3:

Overall, this is a comprehensive and well written manuscript. A high-quality chlorella genome will be useful to the community.

Despite extensive data assembly and analysis, the manuscript could be improved. Specifically, when increased biomass is reported this should be further quantified in terms of lipids, carbohydrates or proteins (or more cells) to better understand how more biomass is being made. Additionally, more information regarding RMI1 is encouraged. A thorough protein map with putative domains is necessary, as is alignments with homologs. Lastly, a more thorough hypothesis regarding RMI1 mechanism is encouraged.

Response: We sincerely appreciate the reviewer's valuable feedback. We believe that these suggestions will significantly contribute to refining our research and uncovering additional interesting findings. Regarding the six algal mutant lines created in this study, we have evaluated their biomass and cell density under high salinity conditions (70‰), where both metrics were consistently elevated across all six lines compared to the wild-type strain. These results not only lend support to our hypothesis that the genes identified in this study enhance salinity tolerance, but also demonstrate the reliability of the methodology employed to identify genes associated with salt tolerance.

Further investigations into metabolite profiling and a more comprehensive characterization of RMI1, including motif prediction, validation, interactions with transcription factors, and the establishment of regulatory networks centered around RMI1, would significantly deepen the scope of our study. A research proposal incorporating these aspects could prove to be quite valuable.

In fact, with the six algal mutant lines and the plant mutants now established, we have begun to explore potential upstream and downstream regulators interacting with the knocked-out genes. We hope to provide additional evidence in future studies that will further support the conclusions presented in this submission and possibly reveal further insights.

Again, thank you and the anonymous reviewers for the constructive comments that have improved our manuscript greatly.

Sincerely yours,

State Key Lab of Marine Resource Utilization in South China Sea, Hainan

Mailing: 58 Renmin Avenue
Haikou, Hainan, China, 570228

Tel: 86-898-66257612
Fax: 86-898-66251258

Yandu LU, Ph.D

Engineering & Research Center of Marine Bioactives & Bioproducts of Hainan Province

Hainan University, Haikou, China

Email: ydlu@hainanu.edu.cn

<http://orcid.org/0000-0002-0136-2252>

References:

1. Jamy, M. et al. Global patterns and rates of habitat transitions across the eukaryotic tree of life. *Nature Ecology & Evolution* **6**, 1458-1470 (2022).
2. Dittami, S.M., Heesch, S., Olsen, J.L. & Collén, J. Transitions between marine and freshwater environments provide new clues about the origins of multicellular plants and algae. *Journal of Phycology* **53**, 731-745 (2017).
3. Hodač, L. et al. Widespread green algae *Chlorella* and *Stichococcus* exhibit polar-temperate and tropical-temperate biogeography. *FEMS Microbiology Ecology* **92** (2016).
4. Eckardt, N.A. The *Chlorella* genome: big surprises from a small package. *The Plant cell* **22**, 2924 (2010).
5. Barati, B., Lim, P.-E., Gan, S.-Y., Poong, S.-W. & Phang, S.-M. Gene expression profile of marine *Chlorella* strains from different latitudes: stress and recovery under elevated temperatures. *Journal of Applied Phycology* **30**, 3121-3130 (2018).
6. Lu, X. et al. Sustainable development of microalgal biotechnology in coastal zone for aquaculture and food. *Science of The Total Environment* **780**, 146369 (2021).
7. Ramachandran, A., McLatchie, S. & Walsh, D.A. A novel freshwater to marine evolutionary transition revealed within *Methylophilaceae* bacteria from the Arctic ocean. *mBio* **12**, e0130621 (2021).
8. Fan, J. et al. Genomic foundation of starch-to-lipid switch in oleaginous *Chlorella* spp. *Plant physiology* **169**, 2444-2461 (2015).
9. Blanc, G. et al. The *Chlorella variabilis* NC64A genome reveals adaptation to photosymbiosis, coevolution with viruses, and cryptic sex. *The Plant cell* **22**, 2943-2955 (2010).
10. Lemieux, C., Turmel, M., Otis, C. & Pombert, J.-F. A streamlined and predominantly diploid genome in the tiny marine green alga *Chloropicon primus*. *Nature Communications* **10**, 4061 (2019).
11. Ou, S., Chen, J. & Jiang, N. Assessing genome assembly quality using the LTR Assembly Index (LAI). *Nucleic Acids Research* **46**, e126-e126 (2018).
12. Noda, Y. et al. Diurnal regulation of SOS pathway and sodium excretion underlying salinity tolerance of *Vigna marina*. *Plant, Cell & Environment* **n/a**.
13. Polle, J.E.W. et al. Genomic adaptations of the green alga *Dunaliella salina* to life under high salinity. *Algal Research* **50**, 101990 (2020).
14. Deneka, D. et al. Allosteric modulation of LRRC8 channels by targeting their cytoplasmic domains. *Nature Communications* **12**, 5435 (2021).

State Key Lab of Marine Resource Utilization in South China Sea, Hainan

Mailing: 58 Renmin Avenue
Haikou, Hainan, China, 570228

Tel: 86-898-66257612
Fax: 86-898-66251258

15. Castro, B. et al. Stress-induced reactive oxygen species compartmentalization, perception and signalling. *Nature Plants* **7**, 403-412 (2021).
16. Wang, S. et al. Roles of E3 ubiquitin ligases in plant responses to abiotic stresses. *International Journal of Molecular Sciences* **23** (2022).
17. Petrovova, M., Tkadlec, J., Dvoracek, L., Streitova, E. & Licha, I. NAD(P)H-hydrate dehydratase- a metabolic repair enzyme and its role in *Bacillus subtilis* stress adaptation. *PLoS One* **9**, e112590 (2014).

**State Key Lab of Marine Resource
Utilization in South China Sea, Hainan**

Mailing: 58 Renmin Avenue
Haikou, Hainan, China, 570228

Tel: 86-898-66257612
Fax: 86-898-66251258

July 4, 2025

Dear Reviewers,

Thank you for the constructive and extremely valuable comments to help improving the manuscript. Here we are submitting a revised version, with item-by-item responses (**bold fonts**) to the comments below.

We have thoroughly proofread and edited the text following the requirements. We believe that our revised version supports the main conclusion of our initial submission and also reveal additional insights, which were detailed below.

Comments from Reviewer 1:

1. The authors of the manuscript entitled "Cross-species dissection of saline-related genes by genetically deciphering a euryhaline microalga *Chlorella* sp." have submitted a detailed revised manuscript addressing the reviewers' comments. I appreciate the extensive effort they have invested in this revision process. However, several modifications are still required before publication:

The methodology section requires significant improvement. Currently, it merely lists section headings without providing adequate experimental details. Authors must develop each methodological section with sufficient information to ensure reproducibility. If the comprehensive methodology would make the main text too lengthy, detailed protocols should be provided as supplementary files. The current format, which presents only section titles without substantive content, is inappropriate for a scientific publication. Please revise this section to include complete methodological descriptions.

Response: We greatly appreciate the reviewer's inputs. As the journal has no word limit for Methods, we have moved the Methods from Supplementary Note 2 to the Main Text. We believe that the methodological section is now with sufficient information to ensure reproducibility.

Comments from Reviewer 2:

I am highly impressed with the completeness of the T2T-level assembly, notably achieving a single gap.

Is there any support for a gap length of 100 bp? The software might arbitrarily insert the round number. Although the quality of the genome with a single gap is not compromised, if a 100bp gap is an artifact, it is appropriate not to mention it.

Response: We fully understand the reviewer's concern. As a matter of fact, the region upstream of the gap (5,000 bp) exhibits complex repetitive structures, such as tandem repeats, representing inherent assembly challenges. Therefore, the length of 100 bp is not the actual size of the gap but represents a small gap where sequence continuity cannot be established. To avoid potential confusion, we revised the previous sentence in Supplementary note 1 (A single gap was present in the assembled genome at the end of chr15 (1,311,674 bp) with a length of 100 bp from 1,286,591 bp to 1,286,690 bp). Now the text reads:

“A single gap was present in the assembled genome at the end of chr15 (1,311,674 bp due to the difficulty in assembling repetitive fragments of the genome)”

2. Fig. 2

Distinctive colors make this figure clear. However, the background colors are in the front, making the characters hard to see. Please exchange the layers' orders.

Response: We have changed the layers' order as advised. Now the figure reads:

Fig. 2. Phylogenetic position of *Chlorella* sp. MEM25. Rooted phylogenetic tree featured 38 Chlorophyta species and representative species from cyanobacteria, Rhodophyta, Glaucophyta, streptophyte algae, and embryophytes. The species tree was deduced using the most closely related genes within single-copy or multi-copy

**State Key Lab of Marine Resource
Utilization in South China Sea, Hainan**

**Mailing: 58 Renmin Avenue
Haikou, Hainan, China, 570228**

**Tel: 86-898-66257612
Fax: 86-898-66251258**

orthogroups (n=199), based on the Species Tree from All Genes (STAG) algorithm. Species from different phyla are indicated in different background colors. Freshwater and seawater species are shown in green and orange dots. Bootstrap with values of 100% or above 70% are indicated in smaller or bigger dots at the branching points, respectively.

Comments from Reviewer 3:

The authors have adequately addressed my concerns.

Response: We sincerely appreciate the reviewer's valuable feedback. We believe that these suggestions will significantly contribute to refining our research and uncovering additional interesting findings.

Again, thank you and the anonymous reviewers for the constructive comments that have improved our manuscript greatly.

Sincerely yours,

Yandu LU, Ph.D & Professor
Engineering & Research Center of Marine Bioactives & Bioproducts of Hainan
Province
Hainan University, Haikou, China
Email: ydlu@hainanu.edu.cn
<http://orcid.org/0000-0002-0136-2252>